eg



# North American Pleistocene Glacial Erosion and Thin Pliocene Regolith Thickness Inferred from Data-Constrained Fully Coupled Ice-Climate-Sediment modelling

Matthew Drew[1,2] and Lev Tarasov[2]

[1]Department of Earth & Environmental Sciences, Dalhousie University, Halifax, Canada
[2]Department of Physics & Physical Oceanography, Memorial University of Newfoundland, St John's, Canada

**Correspondence:** Matthew Drew (mdrew@mun.ca)

**Abstract.** Beyond the impact of glacial isostatic adjustment, landscape evolution is typically neglected at large scale when considering the basal boundary condition for ice sheet and climate modelling over past glacial cycles. Erosion and changing sediment loads impact bed elevation, land/sea mask, and basal drag. To date, how the above affects past ice sheet evolution is unclear.

Constraining the role that Pliocene regolith may have played in Pleistocene glacial cycle variability requires bounds on the amount of regolith preceding those glacial cycles. However, quantitative bounds on regolith thickness at the spatial scale of the glaciated regions of North America are currently absent in the literature.

To address the above, we present an updated sediment production and transport model with dynamically calculated soft sediment mask, isostatic adjustment to dynamical sediment load and bedrock erosion, and a new subglacial hydrology model

coupled to the Glacial Systems Model. The coupled model is capable of multi-million year integrations driven only by greenhouse gas concentration and insolation. The model passes a set of verification tests and conserves mass. We assess parametric sensitivity of sediment transport rates.

We compare the final sediment solutions in an ensemble of whole-Pleistocene simulations for a range of initial (Pliocene) regolith thicknesses against multiple estimates for present day drift thickness distribution, Quaternary sediment volume in

the Atlantic Ocean, and erosion depth estimates. Consistency of modelling and data constraints requires a Pliocene regolith thickness of less than 50 m.



# 1 Introduction

During the last three million years, the landscape of North America has been heavily influenced by glacial conditions. A confident determination of the amount of glacial erosion that had occurred over this time interval would strongly aid in deciphering this influence. However, while there are observationally-based estimates for the total amount of sediment removed from North America during this time interval, it is largely unknown what fraction of this material was sourced from glacial erosion of bedrock versus removal of pre-existing Pliocene regolith. Furthermore, it is difficult to infer large scale erosion rates from these estimates as the full area of the catchment (as well as the relative weighting of sourcing within) is generally not well constrained.

The mean Pliocene regolith thickness over previously glaciated regions in North America is unknown – broad scale estimates in the literature are based on sparse data (50 m, Clark and Pollard, 1998) or dubious physical reasoning ($> 100$ m, Willeit et al., 2019). Multiple drill cores in Minnesota show 70 to 184 m of saprolith (in situ, extensively chemically weathered and disintegrated bedrock) underlying glacial deposits Setterholm and Morey (1995). At glacial-sediment-catchment scale, burial dating of early Pleistocene glacial tills indicates the Pliocene regolith must have been thin enough to be removed by the earliest glaciations as later tills had low cosmogenic nuclide concentrations (Balco and Rovey, 2010). Outside glaciated regions, regolith thickness in Australia (sparse glaciation above 1800 m a.s.l., Colhoun and Barrows, 2011) is up to hundreds of metres thick (Wilford et al., 2016). As chemical weathering rates correlate with precipitation (Lebedeva et al., 2010), one could reasonably expect the drier Australian continent to give a lower bound for the amount of regolith overlying the warmer, wetter North American continent in the Pliocene (Brigham-Grette et al., 2013). This conjecture, however, neglects the many other processes driving chemical weathering and chemical weathering rates may decline poleward due to declining temperatures (**?**).

Constraining long term erosion rates is necessary for assessing the likelihood of exhumation of long term nuclear waste repositories by glacial action and thus site selection (McKinley and Chapman, 2009). Capable models of sediment transport history allow for more targeted mineral exploration (Klassen and Gubins, 1997) and potentially lowered environmental impact from such campaigns. From an earth system dynamics perspective, such constraint would permit clearer assessment of the role soft sediment processes may have played in the mid-Pleistocene Transition (the Regolith Hypothesis, Clark and Pollard, 1998). To address this, herein we use data-constrained large ensemble modelling to determine such constraints.

Below, we start with an overview of the state of ice sheet sediment modelling and the model used in these experiments, identifying newly added couplings. We then present model verification tests and sensitivity analysis of the sediment transport component. Following these, we describe the available model constraints, inputs, and experimental results and terminate with a discussion and conclusions.

Terminology can prove misleading in inter-disciplinary settings. Throughout the text we take erosion to mean the process of removing material, i.e. transport. We use weathering to mean the physical weathering or disintegration of intact bedrock into unconsolidated sediment, i.e. production, via quarrying and abrasion. When this is done by chemical dissolution we refer to this as chemical weathering.



## 1.1 Current Models of Glacial Sedimentary Processes

Though current paleo ice sheet models can incorporate relevant glaciological processes, climate representation is the largest challenge for long term earth system simulation. For the applicable time scales, full coupling with a general circulation model (GCM) is not feasible. Paleo ice sheet modelling studies therefore make simplifications such as snap shotting output from a GCM, glacial indexing, using lower fidelity earth systems models of intermediate complexity (EMICs), or even simpler energy

balance models (EBMs).

The challenge of climate representation is further exacerbated by the uncertainty of atmospheric composition extending back to the beginning of the Pleistocene (or earlier in the case of Pollard and DeConto (2020)). Reliable globally averaged estimates of $CO_2$ extend back to 800 kyr in the Antarctic ice core record (Bereiter et al., 2015). While the boron isotope proxy record is extending inferences of atmospheric $pCO_2$ back into the Pleistocene (Chalk et al., 2017), early Pleistocene estimates remain

sparse and uncertain (Hönisch et al., 2009).

Glacial sedimentary numerical models have been around for decades (Harbor et al., 1988; Clark and Pollard, 1998; Pollard and Deconto, 2009; Melanson et al., 2013; Ugelvig and Egholm, 2018). However, they have not received the same attention as other aspects of the glacial ice/climate/earth system. For example, there has not been a model intercomparison project as there have been for other important ice sheet processes. These sedimentary models are essentially unconstrained owing to the

paucity of quantitative constraint available at the continental scale. The majority of subglacial sediment modelling studies have focused on reproducing ice-catchment scale landscape evolution (*e.g.* (Harbor et al., 1988; Braun et al., 1999; Ugelvig et al., 2016)). Only a scant subset of models have been applied to the scale of continental ice sheets (shown in Table 1).

Numerical continental scale ice sheet models have incorporated subsets of the processes relevant to glacial erosion and at a range of complexities (table 1). At first order, a model must simulate the evolution of ice mass and motion combined with

a relationship between motion and sediment transport. Table 1 shows the steady progression of increasing completeness of glacial erosion models over time. Each of these studies must assume initial sediment distribution at simulation start because this quantity is unknown in each case – a further challenge for modelling sedimentary processes.

Simulating removal of regolith along a 1-D flowline profile of the North American ice sheet over the Pleistocene, Clark and Pollard (1998) started with a uniform thickness 50 m of sediment and modelled transport by deformation. They also assumed

the entire base of the ice sheet was warm and that effective pressure was close to zero. Clark and Pollard (1998) had to cap ice free areas at 100 m sediment thickness to avoid accumulating around 500 m of sediment at the Southern margin, citing missing non-glacial transport as a short-coming.

Pollard and DeConto (2003) use the same transport model as in Clark and Pollard (1998), supplemented with a sediment production law and applied to a 2-D ice sheet model in Antarctica over 400 kyr. The ice sheet model is asynchronously coupled

to a GCM with slab ocean with sufficient temporal resolution to capture obliquity and eccentricity cycles. The model neglects subglacial hydrological processes simulating basal water pressure, thereby requiring the same effective pressure assumption as Clark and Pollard (1998): effective pressure in the sediment layer is only due to sediment weight (water pressure is set = ice sheet overburden). These simulations demonstrate the ability of ice basal action to move tens of metres of sediment from large



| Model Reference | Production | Transp | Hyd | Therm | IA | Ice Dyn | Slid Cpl | Dim | Scale |
|---|---|---|---|---|---|---|---|---|---|
| Jenson et al. (1996) | None | Sub | N | N | ice | SIA | N | 1D | 1500 km/60 kyr |
| Clark and Pollard (1998) | None | Sub | N | N | ice | SIA | N | 1D | 2500 km/3 Myr |
| Dowdeswell and Siegert (1999) | Emp | Emp | N | N | N | SIA | N | 2D | |
| Pollard and DeConto (2003) | Emp | Sub | N | Y | ice | SIA | N | 2D | 4000 km/400 kyr |
| Hildes et al. (2004) | Abr/Qr | Eng | N | Y | ice | SIA | N | 2D | 3500 km/120 kyr |
| Melanson et al. (2013) | Abr/Qr/Emp | Eng/Sub | Y | Y | ice | SIA | N | 2D | 3500 km/120 kyr |
| Pollard and DeConto (2020) | Emp | Sub | N | Y | ice/sed | Hyb | Sed | 2D | 4000 km/40 Myr |
| This work | Abr/Qr | Eng/Sub | Y | Y | ice/sed | Hyb | Sed/Hyd | 2D | 3500 km/2.5 Myr |

**Table 1.** Summary of processes applied in sediment models in the literature. Production is the type of sediment production (Emp is an empirical relation with sliding velocity, Abr is abrasion, Qr is quarrying), Transp is the mode of sediment transport (Sub is subglacial soft sediment deformation, Eng is englacial entrainment and advection), Hyd is whether subglacial hydrology is included, Therm is whether thermodynamics is coupled, IA is the component included in the isostatic adjustment loading (sed refers to the inclusion of sediment loading), Ice Dyn is the ice dynamics used (SIA is shallow ice approximation, Hyb is shallow ice/shallow shelf), Slid Cpl is whether the sediment (Sed) and subglacial hydrology (Hyd) components are coupled into the sliding velocity calculation, Dim is model dimensionality (1D for flowline, 2D for plan view), Scale is the length and spatial scale of the model application. In each case the component complexity is not examined and some components in some models have a more complete treatment than others (e.g. the isostatic adjustment model of Pollard and DeConto (2020) is a local isostatic relaxation model versus the model of Melanson et al. (2013) and this work which is a gravitationally self-consistent global visco-elastic model).



portions of the Antarctic continent in areas which are repeatedly warm based. Transported sediment yields deposits hundreds
of metres thick (up to 900 m) after 400 kyr. Interestingly, the authors show large areas which never yield warm based conditions
during the 400 kyr simulation suggesting that pre-glacial sediment may have survived for millions of years.

Pollard and DeConto (2020) extend their effort in Pollard and DeConto (2003) to a continuous simulation over the last 40
Myr. This integration includes improved ice physics and a comparison against seismic observation. Such long model integration
requires low spatial resolution at 80/160 km. The results show sediment production amounts varying 3 orders of magnitude
with some channels incised down to 7000 m. Thickness of offshore deposits reach 10 km on the shelf and may be limited by
their imposed surface slope restriction (0.0075 m/m) simulating slumping. They iteratively model pre-glacial topography and
effectively assume 20 m of sediment over the continent and 100 m over marine areas.

Hildes et al. (2004) model sediment movement and generation over the last glacial cycle with a more comprehensive model
(including erosive processes). They assume that the last interglacial sediment distribution is the same as present. Melanson
et al. (2013) uses a similar model, integrating ice sheet evolution and sediment processes over the last glacial cycle as well as
assuming present day sediment distribution and topography for last interglacial. Melanson et al. (2013) used the estimates of
total Pleistocene erosion from Bell and Laine (1985) as well as transport distances from the Hudson bay.

Here we present a fully coupled model capable of multi-million year integrations which permits long term simulation of
sedimentary processes such that the results lend themselves to comparison against both the Quaternary sedimentary record
and the present day distribution of sediment. For the first time, this model incorporates a fully coupled suite of ice dynamics,
climate, subglacial hydrology and sediment, where the solid earth model incorporates changes in sediment/bedrock load and
the basal velocities depend on the subglacial water pressure and dynamic soft/hard sliding bed mask.

## 2   Model of Glacial Sedimentation

The sediment model (Melanson, 2012) applied in the Glacial Systems Model (GSM) is designed after that of Hildes (2001).
Both employ separate descriptions of abrasion and quarrying for the sediment production part, although Melanson (2012)
provides an optional empirical sediment production as well as a choice between Boulton and Hallet style abrasion whereas
Hildes (2001) uses only Hallet. Hildes (2001) tracks relative hardness of substrate and abrasive, abrasion ceases when the latter
is less than the former, whereas Melanson (2012) assumes the abrading particle is always harder than the substrate. Hildes
(2001) also allows a spatially varying lithology-based calculation of crack growth, and hence of quarrying rates. Hildes et al.
(2004) and Melanson (2012) both use estimates of cavitation (from basal water pressure) in the lee side of basal protrusions to
estimate stress regimes and thus quarrying rates. Both models give englacial transport rates through advection, however, Hildes
(2001) omits subglacial transport.

The GSM sediment model equations are detailed in Melanson et al. (2013) but those details are repeated in appendix B for
convenience. Several aspects of this model have been updated since the publication of Melanson et al. (2013): performance
improvements by shrinking of subgrid arrays (justified by sensitivity analysis) and array vectorization, improved mass con-
servation through synchronizing all model components in a single time iteration loop, and change to generalized coordinate



reference system on an Arakawa C grid. The sediment model is now coupled to the glacio-isostatic adjustment model of the GSM. The coupling accounts for changing surface load from sediment transport and resultant changes in bed surface elevation (affecting ocean and lake loading). The sediment distribution also dynamically sets the hard/soft bed mask for the basal drag
in the ice dynamics solution.

## 2.1 The Glacial Systems Model

The GSM incorporates a fully coupled system of hybrid SIA/SSA ice physics (Pollard and DeConto, 2012) and 3D ice thermodynamics with local 1D permafrost-resolving bed thermodynamics (Tarasov and Peltier, 2007). We use the grounding line flux approximation of Schoof (2007), subgrid surface mass-balance and ice flow hypsometry parametrization of Le Morzadec
et al. (2015), subshelf melt of Lazeroms et al. (2018), surface drainage solver of (Tarasov and Peltier, 2006), and viscoelastic solid earth (Tarasov and Peltier, 1997).

The coupled climate is an ensemble parameter controlled weighting of temperature and precipitation fields from two components: an EBM with 2D geography, non-linear snow and ice albedo feedbacks, and a slab ocean, and a glacial indexing of a set of empirical orthogonal functions (EOFs) extracted from the PMIP I, II, and III modelling results. These modelling projects
used multiple ice sheet boundary conditions – ICE4G, ICE5G, ICE6G, Glac1D, and ANU. Of these, ICE4G did not have a major Keewatin Ice Dome. The glacial indexing (interpolation between present day and last glacial maximum conditions) of the EOFs relies both on the height of the simulated Keewatin ice dome from the ice dynamical model and the temperature of the North Atlantic from the EBM slab ocean model. The basal hydrology model is a linked-cavity drainage system coupled to an efficient drainage tunnel solver (Drew and Tarasov, 2023). A more detailed description of the GSM is forthcoming in
Tarasov, Hank, and LeCavalier (in prep.).

## 2.2 Sediment-Isostasy Coupling

We incorporated changes in surface sediment load and bed rock erosion into the loading changes of the solid earth (Tarasov and Peltier, 2004). The current model time to PD bed elevation changes are given by:

$$\mathrm{h}_b^t - \mathrm{h}_b^{PD} = h_{sed}^t - h_{sed}^{PD} - \mathrm{Ero} + \mathrm{IA}\left(\mathrm{Ero}, h_{sed}^t - h_{sed}^{PD}\right) \tag{1}$$

where $\mathrm{h}_b^*$ are the bed elevations at current time step and PD, $h_{sed}^*$ are the unconsolidated sediment thicknesses, Ero is the total bedrock thickness which is physically weathered into sediment from quarrying and abrasion over the Pleistocene, and IA is the long term isostatic adjustment due to all of these terms (neglecting shorter term ice and water distribution changes). This isostatic adjustment is calculated on the basis of load changes from the equilibrated (present day plus residual GIA, Tarasov et al., 2012) bed:

$$L_{\mathrm{sed+eros}} = \left(h_{sed}^{PD} - h_{sed}^t\right)\frac{\rho_{sed}}{\rho_{ice}} - \mathrm{Ero}\frac{\rho_{rock}}{\rho_{ice}} \tag{2}$$

In fig. D1 we add varying thicknesses of sediment to show the impact on elevation changes of the solid earth surface (not including sediment) due to isostatic adjustment.





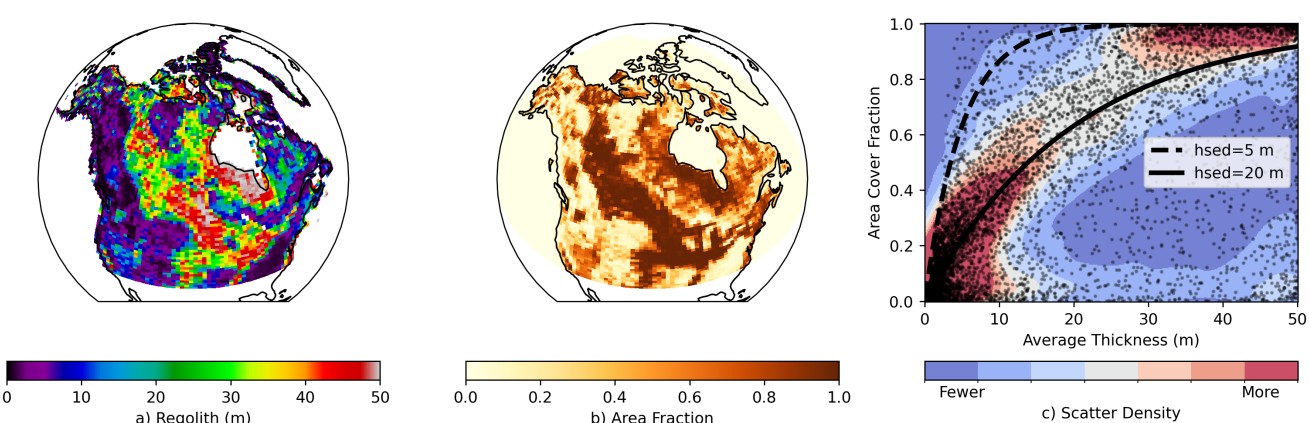

**Figure 1.** Comparison of present day regolith thickness and area based sediment covered fraction from upscaling the regolith field of Pelletier et al. (2016). Distribution of the upscaled thickness-cover fraction points used for developing a relationship between modelled sediment thickness and soft bed fraction.

## 2.3 Basal Drag Coupling

The basal velocity is from either a hard or soft bed sliding rule. For the hard bed the basal sliding rule is a fourth power
Weertman sliding law (Cuffey and Paterson, 2010):

$$u_b^{hard} = \frac{c_{hard} f_{Neff} |\tau_b|^3 \tau_b}{N_{eff}} \tag{3}$$

where $c_{hard}$ is a parameterized sliding coefficient which includes a parameterization for basal roughness, $f_{Neff}$ is the effective pressure normalization factor, $N_{eff}$ is the effective pressure given by the subglacial hydrology model, and $\tau_b$ is the basal drag. The basal velocity for soft bedded sliding is similarly a Weertman type sliding law with a parameterized power with integer
values between one and seven.

$$u_b^{soft} = \frac{c_{soft} f_{Neff} |\tau_b|^4 \tau_b}{N_{eff}} \tag{4}$$

with separately parameterized soft sliding coefficient $c_{soft}$ (which also includes a parameterization for basal roughness).

For a soft bed fraction greater than one half, the sliding rule is a soft bed rule, otherwise it is hard. The soft bed fraction is calculated with a parameterized relationship with sediment thickness ($h_{sed}$). To develop this relationship the high resolution
(1 km) sediment dataset of Pelletier et al. (2016) (shown in fig. 1 b)) was upscaled to the GSM North American $1.0 \times 0.5°$ latitude-longitude grid. The distribution of points in fig. 1 was used to derive a parameterized empirical relationship between sediment thickness and soft bed fraction ($f_{soft}$) to estimate the shielding factor, $h_c$:

$$f_{soft} = 1 - e^{h_{sed}/h_c} \tag{5}$$

This shielding factor is also used in the quarrying (eqn. B12), abrasion (eqn. ), and englacial sediment mixing coefficient
(eqn. **??**) relationships.



## 2.4 Sediment Model Verification Tests

To verify that the model solves the physical equations in § B as intended, a series of model verification tests were performed. The verification strategy was three-fold: symmetric solution given symmetric boundary conditions, convergence to a continuous solution given increasing grid resolution, and conservation of mass. The test bed used for these model solutions was a square

root ice sheet topography with coupled subglacial hydrology, sediment, and dynamically calculated basal sliding velocity (from the effective pressure solution of the subglacial hydrology and the driving stress from the square root ice surface).

## 2.5 Sediment Transport Sensitivity for a Square Root Ice Sheet

| Symbol | Description | Distribution | Justification |
|---|---|---|---|
| $K_{cond}$ | hydraulic conductivity of cavity network | $[1e-4, 3e-1]$ | (Drew and Tarasov, 2023) |
| $h_r$ | basal protrusion height | $[0.01, 20]m$ | (Drew and Tarasov, 2023) |
| $l_r$ | basal protrusion wavelength | $[10, 20]m$ | (Drew and Tarasov, 2023) |
| $T_{crit}$ | basal freeze point (PMP rel.) | $[-1, 0]$ | Ansatz |
| $f_{N_{eff}}$ | $N_{eff}$ basal sliding term | $[1e4, 2.5e7]$ | (Drew and Tarasov, 2023) |
| $HV^\star$ | Vicker hardness bedrock | $[1.5e9, 7e9]$ | (Hildes, 2001) |
| $\mu^\star$ | Rock-rock friction coef. | $[0.3, 0.85, 10]$ | (Byerlee, 1978) |
| $R^\star_{mean}$ | Mean size of grain dist. | $[10^{-4}, 10^{-3}, 10^{-2}]m$ | (Boulton, 1979; Haldorsen, 2008; Hubbard and Sharp, 1995) |
| $h^\star_f$ | water film thickness | $[10^{-8}, 10^{-6}, 10^{-4}]m$ | (Hubbard and Nienow, 1997; Hallet, 1979) |
| $C^\star_{quar}$ | quarrying coefficient | $[10^{-11}, 10^{-10}, 10^{-9}]Pa^{-1}$ | Ansatz, $\propto k_{abr}/HV^{star}$ |
| $\zeta^\star_b$ | basal roughness | $[0.01, 0.1, 1]$ | $\propto$ size erratics (Krabbendam and Glasser, 2011), obstacle size (Weertman, 1957) |
| $n^\star_p$ | quarrying law exponent | $[0.3, 0.5, 1]$ | (Cuffey and Paterson, 2010) |
| $h^\star_{sed}$ | shielding factor | $[2, 6, 20]m$ | analysis of Fig. 1 |
| $\phi^\star$ | internal friction angle basal sed. | $[20, 22, 24]°$ | (Jenson et al., 1995) |
| $n^\star_s$ | rheology exponent of basal sed. | $[1.25, 1.5, 1.75]$ | (Jenson et al., 1995) |
| $\mu^\star_0$ | Newtownian reference viscosity basal sed. | $[0.1, 5, 100] \times 10^9 Pas$ | (Jenson et al., 1995) |
| $\phi^\star_p$ | sediment porosity | $[0.3, 0.4, 0.5]$ | Ansatz |
| $Z^\star_r$ | thermal resistivity of englacial debris | $[0.23, 0.55, 1.1]mK/W$ | (Hildes, 2001; Hallet, 1979) |
| $C^\star_{max}$ | upper bound on sed concentration by vol | $[.85, 0.9, 0.95]$ | Ansatz |





| $\tilde{D}^{\star}$ | prefactor diffusion coefficient | $10^{[-11,-10,-9]}m^2/s$ | (Alley and MacAyeal, 1994) |
| $\tilde{z}^{\star}$ | scale factor for exponential coefficient in $D$ | $[3,5,7]m$ | Heuristic |
| $C^{\star}_{crit}$ | critical debris concentration | $[0.3, 0.4, 0.5]$ | (Hildes, 2001) |

Table 2: Parameters in the GSM sediment model. Those probed with the transport sensitivity analysis in § 2.5 are highlighted in green.

Sensitivity analysis of model behaviour based on uncertainty in the input parameters identifies those parameters which produce the largest variance in the model response phase space. Those producing the widest range in model response would most effectively reduce uncertainty in system behaviour if the parameter uncertainty were reduced. The process of identifying the most important parameters in this sense is called "factor prioritization" (Saltelli et al., 2008). Analyzing model sensitivity to parameter uncertainty also allows identifying those parameters which produce the least amount of variability and, due to a scarcity of computational resources, are not worth probing. The process of identifying those parameters which may be held constant when exploring model response in order to capture system behaviour is called "factor fixing" (Saltelli et al., 2008). In this section we aim to identify those parameters the coupled subglacial hydrology-sediment system which may be fixed and those which are the most important controls on sediment transport.

For most systems, conducting grid search for a 15-dimensional parameter space is unfeasible. Though it depends on model response surface rugosity, a grid search is likely not necessary and low discrepancy sampling methods have been demonstrated to capture phase space variability for some non-linear systems (e.g. see Saltelli (2002)). The most commonly applied sensitivity analyses are variance based methods.

We focus here on sediment transport sensitivity, the parameters controlling sediment production rates are known well enough a priori. Here we use both a variance and a distribution based methods to capture model sensitivity and assess the relative importance of various parameters. These sensitivities are used for selecting ensemble parameters in the following sections and setting the rest to constant values.

Variance based sensitivity indices ($S_i$) of model response ($y$) sensitivity to model input ($x_i$) are built on conditional variances (Saltelli et al., 2008):

$$S_i = \frac{V_{x_i}\left(E_{x\ i}(y|x_i)\right)}{V(y)} \tag{6}$$

which Saltelli et al. (2019) describes as "the expected fractional reduction in the variance of y that would be achieved if factor $x_i$ could be fixed. $S_i = 1$ implies that all of the variance of $y$ is driven by $x_i$, and hence that fixing it also uniquely determines $y$."

Distribution based sensitivity indices are based on a measure of change in the model response distribution from fixing $x_i$ – the difference between the unconditional probability distribution of $y$ and the distribution conditional on a given value of $x_i$. The Kolmogorov-Smirnov statistic is commonly used as a non-parametric measure of the discrepancy between two distributions





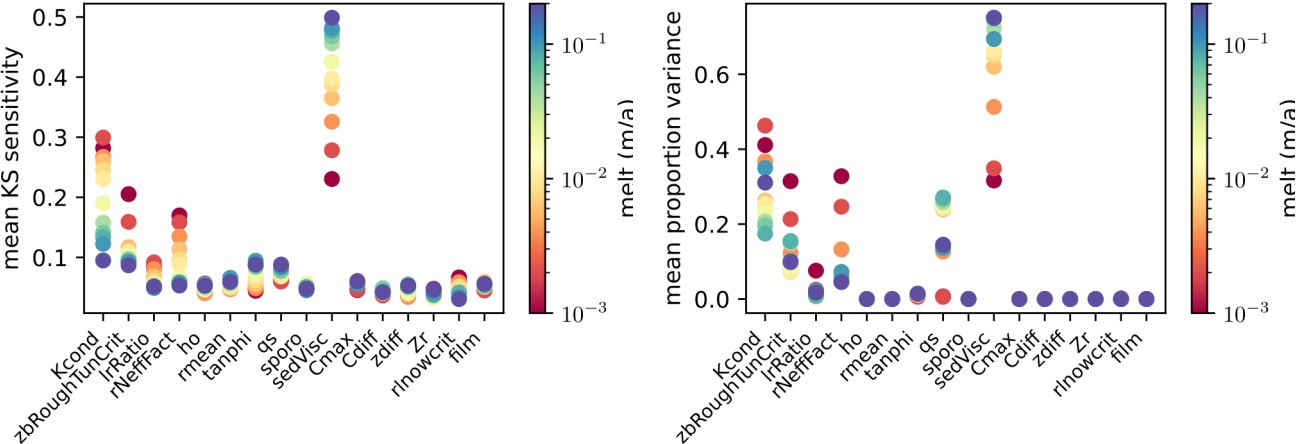

**Figure 2.** Sediment transport sensitivity to parameters. Mean KS sensitivity is the metric of Pianosi and Wagener (2015), mean proportion variance is the metric of Saltelli et al. (2008).

which Pianosi and Wagener (2015) use to measure the conditional-unconditional difference:

$$S_i = \underset{x_i}{median}\left(KS(x_i)\right), \text{ where } KS(x_i) = \underset{y}{max}\left|F(y) - F(y|x_i)\right| \tag{7}$$

where $F(y)$ is the unconditional cumulative distribution function of the model response and $F(y|x_i)$ is the cumulative distribution function which is conditional on selecting $x = x_i$.

For both sensitivity metrics, the amount of sediment transported is most sensitive to the sediment viscosity (fig. 2). The next most important are subglacial hydrology parameters. We therefore assign the subglacial hydrology parameters, sediment viscosity, and the shielding factor ($h_c$, ho in fig. 2 to which basal drag and erosion are sensitive) to the ensemble parameter set and assign constant median values to the remaining subglacial sediment model parameters.

## 3   Constraints

### 3.1   Present Day Sediment Distribution

To answer what constraint is offered by glaciological constraint and the present day distribution of unconsolidated sediment on pre-glacial sediment thickness, we must first have an estimate for the present day sediment thickness. As uncertain as the Pliocene unconsolidated sediment distribution is, even the present day sediment distribution has large uncertainty. The sediment distribution is most uncertain in Canada where data is sparse. In the United States, relatively dense population gives a wealth of well data whereas broad regions of Canada are sparsely populated. There are some exceptions to this. In Southern Ontario and Quebec, well data, geological surveys and high resolution DEMs give more reliable estimates. Some of the most important areas from a glaciation perspective (*e.g.* Northern Saskatchewan and Manitoba, Northwest Territories and Nunavut,



| Ref. | Region | Source Data |
|---|---|---|
| (Taylor, 2023) | Glaciated N. Am. | Geol |
| (Soller and Garrity, 2018) | Glaciated US East of Rockies | Mult. USGS Databases |
| (Pelletier et al., 2016) | Global Terrestrial | Wells, Machine Learning |
| (Shangguan et al., 2017) | Global Terrestrial | Wells, Machine Learning |
| (Straume et al., 2019) | Global Marine | Seismic & Borehole |
| (Laske et al., 2013) | Global Terrestrial & Marine | Seismic |

**Table 3.** A summary of large scale datasets available, their geographical coverage, and data sources.

Northern Quebec-Labrador) have no constraint beyond surficial geology mapping (Fulton, 1995) and digital elevation maps from remote sensing (Porter et al., 2018). As such the sediment thickness estimates in these regions are transform functions of those products. Some seismic data are available for reconstructing sediment thickness offshore. However seismic data collection in areas with sea ice cover is difficult and data in the Arctic are sparse (Stashin, 2021).

Soller and Garrity (2018) produced a composite map of sediment thickness and bedrock elevation for the glaciated regions of the Northern United States to the East of the Rocky Mountains from many smaller scale regional maps (in fig. 3 d). Naylor et al. (2021) constructed a bedrock elevation map in order to assess present day sediment thickness and bedrock erosion during the Pleistocene. This product synthesized regional maps from many states and provinces in the United States and Canada, subsequently validating the final result against available well data.

The sediment distribution by the Canada1Water initiative brought together maps by Soller and Garrity (2018), Parent et al. (2021), Russell et al. (2017), and Smith and Lesk-Winfield (2010). Areas outside these maps were assigned thicknesses based on surficial geologic units from Sur (2014), Karlstrom (1964), and GIS Team National Park Service (1999) with assumed thicknesses for various geologic units in as laid out in table A1.

The offshore areas of Laske et al. (2013) and Straume et al. (2019) are remarkably similar. Indeed, both datasets use seismic
methods to estimate the depth of unconsolidated sediment on the oceans seafloors and provide excellent coverage. The issues with using these data to constrain modelled sediment fields are two fold: it is unclear what proportion of those depths are Quaternary aged sediment and it is unclear whether the reported layers which are based on seismic p-wave thresholds correspond to unconsolidated sediment or other units.

Onshore, there is broad agreement on thick sediment coverage in the southern prairies and north-central US and thin sediment
over the Rockies. Interestingly, only (Pelletier et al., 2016) gives thick sediment cover of the Hudson Bay Lowlands (south of Hudson Bay). In the Canadian Arctic Archipelago, (Shangguan et al., 2017) and (Pelletier et al., 2016) both show thick sediment over Banks and Victoria Island and moderate sediment over Baffin while (Taylor, 2023) shows thin sediment over the region north of the Arctic Circle as a whole. In terms of total sediment volume, the three reconstructions vary by more than 100% between the most and least capacious: Pelletier et al. (2016) gives 310000 km$^3$, Shangguan et al. (2017) gives 460000
km$^3$, and Taylor (2023) gives 200000 km$^3$.

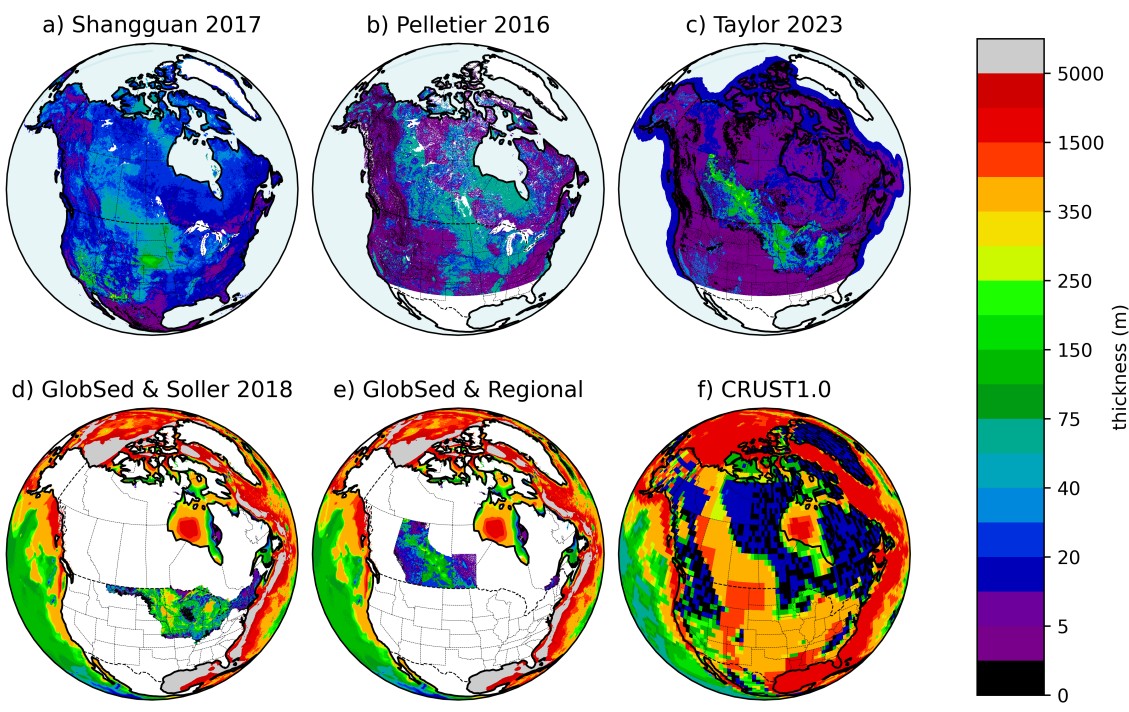

**Figure 3.** Comparison of various present day unconsolidated surface sediment thicknesses. a) dataset of Shangguan et al. (2017) produced with machine learning methods, b) dataset of Pelletier et al. (2016) produced with empirical relationships from soil profiles and water wells, c) dataset of Taylor (2023) which incorporates many data sources, d) dataset of Soller and Garrity (2018) for the onshore and Straume et al. (2019) for the offshore, e) regional datasets of (from west to east) Atkinson et al. (2020); Fenton et al. (1994); Keller and Matile (2021); Parent et al. (2021) for the onshore and Straume et al. (2019) for the offshore, and f) dataset of Laske et al. (2013). The three datasets (Pelletier et al., 2016; Shangguan et al., 2017; Taylor, 2023) with the best terrestrial coverage were selected for constraining the spatial distribution in the model results. The regional datasets are compared below with bulk volumes for those areas.




Though nearly forty years old, the estimate of Quaternary aged material deposited in the oceans around the domain of the Laurentide ice sheet by Bell and Laine (1985) is still the most comprehensive to date. They detail estimates for the Gulf of Mexico, the west Atlantic, and the Canadian Arctic on the basis of drilling and seismic surveys. Due to the harsh and sensitive environment of the Arctic both drilling and seismic are difficult even today. As such their calculation of Quaternary material for this region has large uncertainties and they state they cannot provide a reliable calculation for this region. Bell and Laine (1985) state that the West Atlantic has excellent seismic coverage and, referring to several studies therein, estimate $1.2 \times 10^6$ km$^3$ of Laurentide sourced Quaternary aged sediment in this region. From multiple data sources Bell and Laine (1985) calculate $7.4 \times 10^5$ km$^3$ of Quaternary sediment in the Gulf of Mexico. Hay et al. (1989) rejected the Bell and Laine (1985) estimate of Laurentide sourced sediment due to a disagreement on the proportion of Mississippi transported sediment of non-glacial origin. Whereas Bell and Laine (1985) estimate this rate on the basis of the Miocene accumulation rate in the Gulf, Hay et al. (1989) point out that the present day sediment load of the Mississippi river is brought by the Missouri River from the Rockies, not the Laurentide region. Furthermore, because there is no fluvial transport in our model, relating the glacial sediment estimate from (Bell and Laine, 1985) to material glacially transported to the southern margin in our simulations is non-trivial. Because of the above uncertainties, we focus on comparing our model results against the Atlantic region estimate.

## 3.2 Erosion Estimates

Naylor et al. (2021) estimate 71 metres of bedrock erosion from all sources since the Pliocene in an area encompassing Southern Ontario, Northern US and parts of the Canadian prairies. Ehlers et al. (2006) use apatite (U-Th)/He to estimate hundreds to thousands of metres of bedrock erosion over the Pleistocene. Gulick et al. (2015) use seismic interpretation to reconstruct Pliocene to Pleistocene offshore sedimentation rates. They infer an increase in erosion across the Plio-Pleistocene Transition (PPT) and again during the mid-Pleistocene Transition (MPT). It is not easy to identify the catchment from which these sediments came and so inferring a catchment wide erosion rate is difficult. Though the unstated uncertainties are likely large, the Naylor et al. (2021) erosion estimate is the most useful constraint for modelling North American glacial erosion over the Pleistocene at a continental scale. The bounds on Cordilleran erosion are wide and likely impose less of a constraint on the processes in such a model, meanwhile it is not clear what the precise source region is for the sediment record interpreted by Gulick et al. (2015).

## 4 Uncertainty in CO$_2$ Forcing

Proxy data for atmospheric CO$_2$ concentration prior to coverage by the ice core record (PD-800 ka, Bereiter et al. (2015)) are sparse with wide uncertainty bounds (Hönisch, 2021). Carbon dioxide is well mixed in the atmosphere and the gas is exchanged quickly on paleo timescales at the sea surface. As such that sea surface carbonate chemistry is sensitive to changes in mean atmospheric CO$_2$. The $^{11}$B isotope proxy is sensitive to changes in pH and as such is used as a proxy for atmospheric carbon dioxide (Hönisch et al., 2009). The benefits of this record are its temporal range, estimates are available back to 2.1 Ma. However, the temporal coverage is sparse and not orbitally well-resolved (e.g. Dyez et al., 2018). As such boron isotope





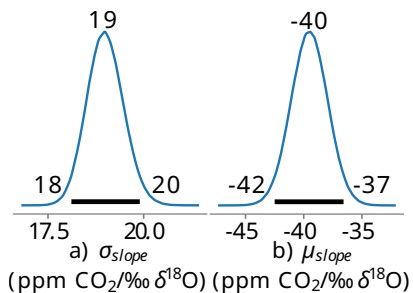

**Figure 4.** Distribution of the mean and standard deviation of the slope of a linear fit from 800-0 ka between ProbStack (Ahn et al., 2017) and EPICA (Bereiter et al., 2015). The mean value was removed from each dataset prior to fitting. Trends and data can be seen in fig. 4.

records are not sufficient for developing a $CO_2$ forcing. Other proxies have poorer coverage and larger uncertainties (Willeit et al., 2015; Hönisch, 2021).

The benthic $\delta^{18}O$ climate proxy record has excellent temporal coverage over the Plio-Pleistocene and is sensitive to changes in global temperature, of which $CO_2$ forms a part of the signal. We adopt the approach of deriving a $CO_2$ forcing based on this $\delta^{18}O$-temperature-$CO_2$ relationship with the boron isotope record as a validation dataset. As a start point we develop an empirical $\delta^{18}O$-$CO_2$ relationship through regression of the European Project for Ice Coring in Antarctica (EPICA) ice core (Bereiter et al., 2015) and ProbStack (an update to the 5 Myr LR04 stack of 57 benthic records incorporating an additional

123 records, better coverage outside the Atlantic Ocean, and improved uncertainty analysis,  Ahn et al., 2017) records. We assume a linear relationship between benthic $\delta^{18}O$ and $CO_2$ as the analysis of Dyez et al. (2018) does not support higher order terms. We use the PyMC python package for probabilistic programming (Abril-Pla et al., 2023) to develop a generalized linear model through Hamiltonian Monte Carlo sampling (specifically No U-Turns Sampler) and obtain distributions for the linear regression coefficients. First the mean was subtracted from both datasets, then the sampler was run on 16 chains with

20000 draws. This yielded distributions for intercept ($\approx 0$, left panel fig. 4), mean slope estimate ($\approx 40$ ppm/‰, centre panel fig. 4), and standard deviation of the slope estimate ($\approx 19$ ppm/‰, right panel fig. 4). A comparison of the underlying data and the final chosen linear relationship along with a comparison of the output time series and the boron isotope record from data in (Dyez et al., 2018) can be seen in fig. 5. The relationship between oxygen isotopes and carbon dioxide is assumed to change sometime around the MPT and so the slope and intercept coefficients are linearly interpolated to those of the maximum

likelihood EPICA-ProbStack linear regression over 1300 to 900 ka. This linear fit persists until EPICA coverage begins at 798 ka. The carbon dioxide time series from the oxygen isotope regression were merged with the carbon dioxide measurements from the EPICA record using a linear weighting over 750 to 798 ka. We test multiple $CO_2$ forcings on the basis of this distribution, finding that a $CO_2/\delta^{18}O$ slope four standard deviations away from the expected slope and a mean $CO_2$ of 266 ppm versus the mean EPICA $CO_2$ of 236 ppm were necessary to attain a target sea level amplitude during the early Pleistocene

similar to that of Elderfield et al. (2012) and Rohling et al. (2014).This significant departure (four standard deviations) from the expected regression values of EPICA vs ProbStack is in line with the findings of Dyez et al. (2018) who found that the ice



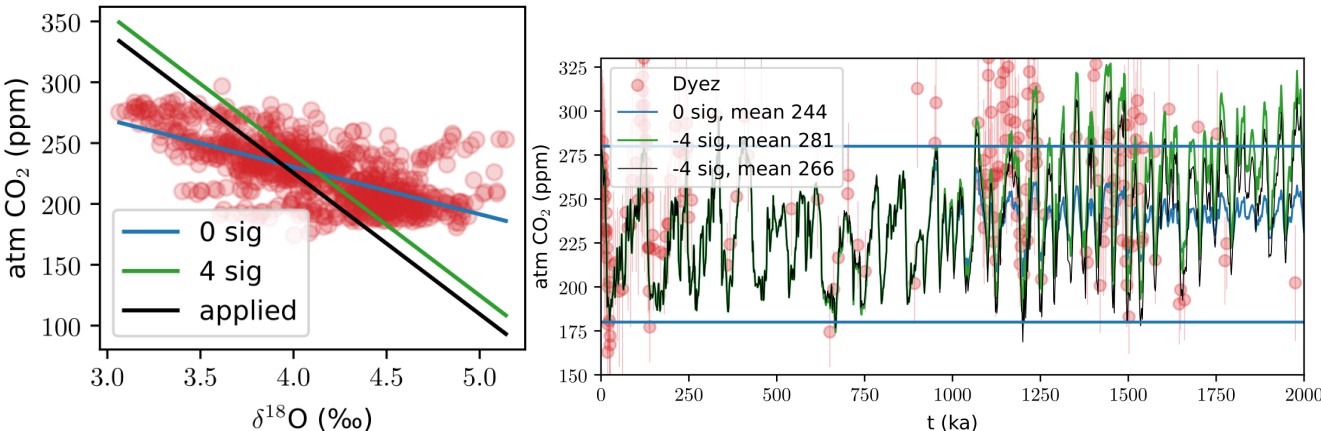

**Figure 5.** Scatterplot and time series showing the fit of relationships from a linear fit over 800-0 ka between ProbStack (Ahn et al., 2017) and EPICA (Bereiter et al., 2015). Various $\sigma$ ("sig" in plot legend) based on HMC results in fig. 4.

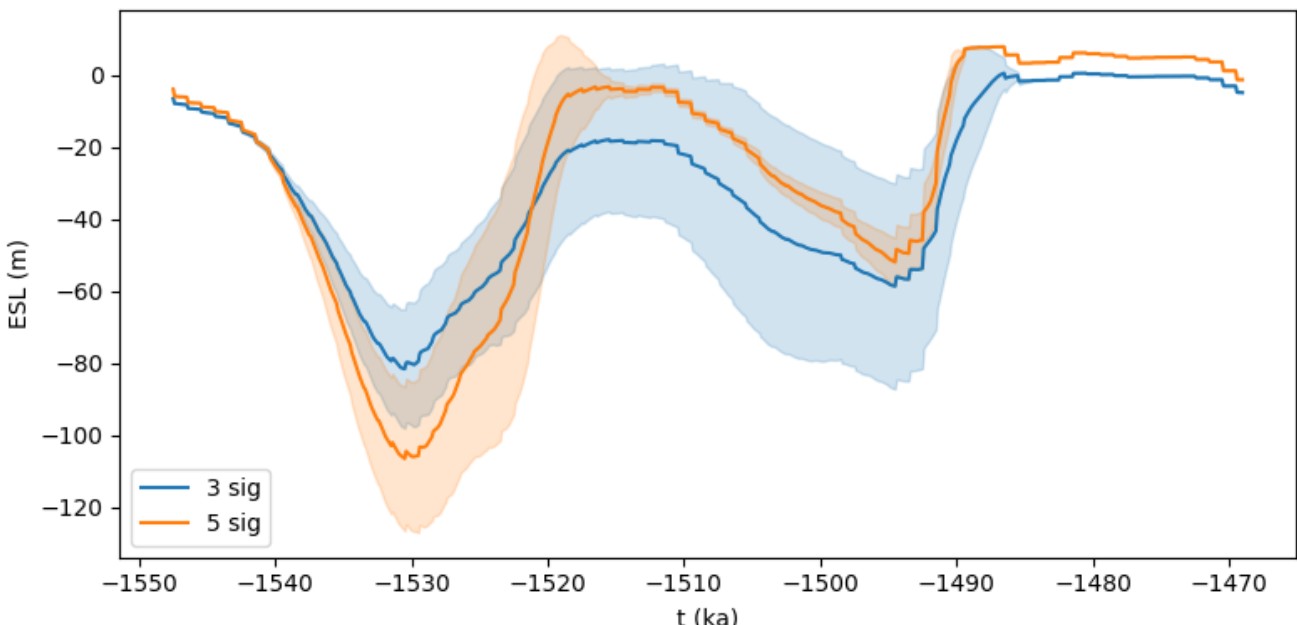

**Figure 6.** Resulting ensemble mean sea level curves (with one standard deviation shading) for 35 parameter vectors run for $CO_2$ from ProbStack using a slope three $\sigma$ and five $\sigma$ away from the expected value from the distribution shown in fig. 5.

volume-carbon dioxide relationship as approximated by the boron and oxygen isotope records changed significantly across the MPT.





## 5 Methodology

The GSM was run for both the North American Ice Complex (the primary target for our sedimentary study) and the Eurasian Ice Complex (to incorporate a stronger obliquity response). To obtain a set of parameter vectors which reproduce a realistic North American Ice Complex, the model was tuned through several iterations of Latin hypercube sampling for GSM ensemble parameters, simulating the last glacial cycle (-122 ka to present day) and multiple early Pleistocene glacial cycles (MIS 55-45, 1603-1400 ka, an interval with low $CO_2$ in our forcing), omitting the subglacial hydrology and sediment model components.

These sample distributions were then sieved (all runs which lie outside the range for each constraint metric were removed) by bounds on inferred last glacial maximum (LGM) sea level (Spratt and Lisiecki, 2016), deglacial margin chronology (Dyke, 2004), Early Pleistocene sea level amplitude (Rohling et al., 2014; Elderfield et al., 2012), and Early Pleistocene extent (Roy et al., 2004; Balco and Rovey, 2010). These sieved distributions were then fit with a beta distribution and re-sampled with a Latin hypercube for 2 more iterations. This gave a set of basis vectors for the ice dynamic and climate parameters which

were then combined with samples for the subglacial hydrology model and re-sieved. This basis vector selection, sampling and combination was done again with the sediment model coupled in.

    Simulations with the sieved ensemble parameter vectors were then run for the full Pleistocene continuously (2.58 Myr) with uniform Pliocene regolith distribution initial conditions of 0-120 m thickness with increments of 10 m over both the North American and Eurasian model domains. This range of Pliocene regolith thicknesses covers the mean material removal over

glaciated North America estimated by Bell and Laine (1985).

## 6 Results

### 6.1 Data-Model Comparison

  Here we compare model results with the present day sediment distribution reconstructions by Taylor (2023), Shangguan et al. (2017), and Pelletier et al. (2016). Each data set was interpolated to the GSM model grid for comparison against model results

(`cdo remapbil`, Schulzweida, 2023). The data of Pelletier et al. (2016) were extrapolated over missing data in lakes and ocean areas then upscaled (`cdo fillmiss` Schulzweida, 2023). Prior to comparison, all three data sets were masked to the same coverage as Shangguan et al. (2017) (no coverage over ocean areas, great lakes and major lakes in the Canadian Prairies). The max and min across the three data sets for each cell was extracted and used as the bounds for misfit scoring. Modelled thicknesses which lie within these maximum and minimum bounds were assigned zero misfit, outside this range the difference

with the closest bound was used. In Fig. 7 a) we examine the mean L1 norm misfit with respect to these bounds.

    Taken as a whole, these results presented in Fig. 7 and further examined in Fig. 8 to incorporate glaciological controls indicate that Pliocene regolith thicknesses were likely less than 50 m. Fig. 7 shows that the match between simulated and observed present day sediment distribution gets progressively worse as the regolith scenarios thicken. This may be the result of underestimating the thickest sediment packages by reconstruction schemes which extrapolate thin sediment coverage outside







**Figure 7.** Comparison of observed present day sediment volumes and erosion thickness estimates against modelled quantities for varying levels of starting sediment thickness (regolith scenarios). The red dots indicate mean and short red lines indicate median for each regolith scenario. Panel a) shows the L1 misfit with the estimated present day distribution of sediment as detailed in sec. 6. In panel b) the Bell and Laine (1985) estimate for total Quaternary sediment deposited from North America into the Atlantic is shown by a horizontal red line. Panel c) shows modelled and observed sediment volume for Alberta by Atkinson et al. (2020), d) for Saskatchewan by Fenton et al. (1994), e) for Manitoba by Keller and Matile (2021), and f) for the St. Lawrence estuary by Parent et al. (2021). Panel g) compares the modelled and observed sediment volume for the formerly glaciated region of the United States East of The Rockies by Soller and Garrity (2018). In panel h) the Naylor et al. (2021) estimate for mean bedrock erosion depth (71 m) in their study area is shown by a horizontal red line. Panel i) is the erosion depths in the study location of Ehlers et al. (2006) where bounds on observed values are 100's to 1000's of metres.



areas of better data coverage on the basis of surficial geology (Taylor, 2023) or change in relationship between input data and sediment thickness further North due to a change in surface processes (Pelletier et al., 2016; Shangguan et al., 2017).

   Regional datasets were collected where available at scales comparable to that of the model. This covered the prairies of Western Canada, the St Lawrence valley and the Atlantic Basin estimate of Bell and Laine (1985). We compare the model results against these regional datasets from west to east. Bulk volume estimates are less prone to error than the individual

locations and so are a better target for data-model comparison. For all regions, the modelled bulk sediment volume increases with starting regolith thickness – though the trend is more linear for some regions than others.

   The amount of sediment exported to the Atlantic basin increases roughly linearly with increasing amounts of pre-glacial regolith. This indicates that, in this large depo-zone for a significant portion of the Laurentide catchment, the North American Ice Complex does not suffer from shielding of bedrock from weathering by regolith cover. The ice sheet is able to evacuate

sufficient sediment such that erosion can continue to produce nearly the same amount of sediment in the thicker regolith scenarios. The mean and median estimates for the 30 and 40 m ensembles in fig. 7 b) agrees remarkably well with the amount of sediment transported to the North American Atlantic sector estimated by Bell and Laine (1985). However this estimate carries large uncertainty and both the 10 m and 50 m ensemble median values lie within 50% uncertainty bounds.

   The Alberta drift thickness dataset is the isopach for the land surface to top of bedrock difference (Atkinson et al., 2020). As

such, errors in the bedrock elevation surface will propagate into the drift thickness dataset. This bedrock elevation surface was produced with machine learning techniques which used training and validation data from boreholes and 39 terrain and geologic input datasets. The sediment volume increases linearly at first but becomes erratic after about 50 m of regolith, possibly due to the dual influence of the Cordilleran and Laurentide ice sheet not present in the other regions. Here the observed bulk sediment volume stays within the interquartile range for all ensembles except the 0 m one. The trend in ensemble median values is

closest to this observed value between 20-40 m of regolith.

   Fenton et al. (1994) reconstructed the drift thickness in Southern Saskatchewan. In this region the modelled sediment volume increases linearly out to about 90 m and then flattens off indicating significant shielding of the bedrock from weathering. The observed bulk sediment volume lies within the modelled interquartile range for starting regolith thicknesses from 10 to 20 m.

   The Southern Manitoba region here appears to be a bit of an outlier among the prairie provinces. Sediment volumes recon-

structed by Keller and Matile (2021) from multiple borehole databases are five times lesser in Southern Manitoba than those of (Fenton et al., 1994) in Southern Saskatchewan (though the areas are comparable). The lower Manitoba sediment volume is not reflected in the model results which give similar volumes in both provinces. The sub-linear trend in sediment volume with increasing regolith thickness indicates that bedrock shielding may be more significant here than in other sectors of the ice sheet. This may result from insufficient transport capacity to denude the bedrock or some significant process absent the model.

In this region none of the regolith scenarios agree with the observation.

   The till thickness reconstruction of Parent et al. (2021) collated geological models for multiple subregions within the study area based on borehole and geophysical data, filling gaps between them. Examining the established stratigraphy in the area Parent et al. (2021) describe multiple glacial advances within the last glacial cycle alone – the Quaternary record in this area has been repeatedly wiped clean. Comparing the modelled sediment thickness in this area with their till volume reflects this – the



ensemble median values largely agree with the observed volume regardless of regolith scenario. The fast flowing ice regime
      here has enough transport capacity to denude the bedrock. Interestingly, this is the only region with appreciable departure
      of mean from median modelled sediment volumes – the distributions get progressively more skewed with thicker regolith
      scenarios.

      Fig. 7 g) compares the modelled sediment with the reconstruction by Soller and Garrity (2018) for the previously glaciated
United States areas east of the Rockies. This reconstruction was based on several geologic databases detailed within their
      publication. Sediment volumes increase roughly linearly out to the 90 m regolith scenario. The 20-50 m regolith scenarios
      have interquartile ranges which envelope the observed sediment volume.

      The bedrock removal reconstruction of Naylor et al. (2021) poses an upper bound on physical weathering of the bed by
      glacial processes as that reconstruction does not separate out chemical, biological, and other physical weathering processes
which contribute to the production of sediments (Goudie and Viles, 2012). The median bedrock erosion values in Fig. 7 h)
      are below this estimate for all regolith scenarios and any trend in the amount of bedrock quarried or abraded is small or non-
      existent. This indicates that for the majority of the Southern Laurentide domain, the ice sheet is able to remove the debris
      produced by quarrying and abrasion and bedrock shielding is not a limiting factor.

      In Fig. 7 i) we show the bedrock removal thickness over the study are of Ehlers et al. (2006). They inferred large erosion
bounds in their study area of 100's to 1000's m and, as such, their bounds do not pose a constraint on these results. The wide
      distribution within each ensemble also shows this spread. Whereas the wide bounds on erosion depth inferred by Ehlers et al.
      (2006) is the result of both methodological uncertainty from thermochronometry and likely wide range in basal thermal regime
      in rough terrain, the distribution in each ensemble is due to parametric uncertainty. This illustrates the challenge of reconciling
      the scales of a typical geologic study area and ice sheet modelling.

### 6.1.1   Regolith Thickness Inferred by Applying Constraint Bounds to Model Runs

      Here we discard runs which lie outside the plausible range of earth systems histories on the basis of the uncertainty bounds
      for glaciological and sedimentary constraints – a process referred to as sieving. The sieve was designed based on sensitivity
      of the inferred Pliocene regolith bounds to each constraint, the plausible uncertainty range for that constraint, and uncertainty
      in modelled earth system evolution due imperfect process representation, initial and boundary conditions, and scale. For ex-
ample, while the timing of the removal of the ice dam between the Hudson Bay and Atlantic ocean is well constrained (Alley
      and Agustsdottir, 2005; Thomas et al., 2007; Matero et al., 2017), models aiming to capture the overall features of the full
      Pleistocene glacial cycles will perform more poorly at the millennial scale. As such the runs outside the (loose) 4.1 to 12.3 kyr
      range were discarded.

      The next constraints applied to the ensembles control for extent of the North American ice sheet: scoring against the deglacial
margin chronologies of Dyke (2004) and Dalton et al. (2020) and the early and late Pleistocene most southerly tills of Roy et al.
      (2004) and Balco and Rovey (2010). For these, runs outperforming the ensemble median margin score and those with early
      Pleistocene minimum latitude of ice between 38 and 43 ° N both before and after 1 Ma were kept. The next set of controls
      was early Pleistocene and LGM sea level. LGM sea level is generally accepted to have been between 125 to 140 m below



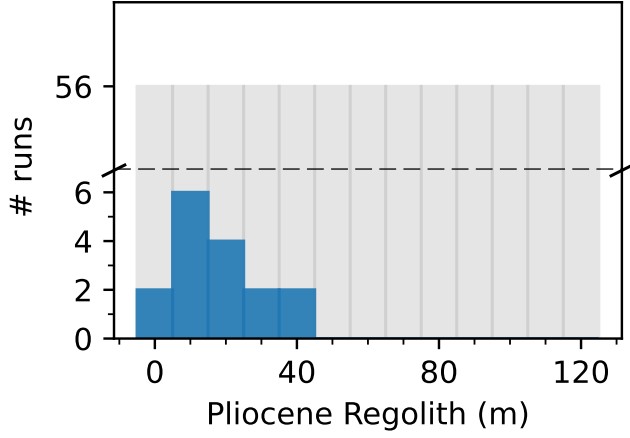

**Figure 8.** Sieved Pliocene regolith thickness distribution (blue) with full ensemble Pliocene regolith thickness distribution (greyed-out, constant 56 runs per level).

present (Austermann et al. (130 m 2013), Lambeck et al. (130-140 m 2014), Yokoyama et al. (125-130 m 2018). Sea level
in the GSM is calculated using a scale factor to convert ice volume to barystatic sea level change and is based on the present day ocean area. Furthermore, only the North American and Eurasian ice sheets are modelled here – for the remaining bodies of terrestrial ice a scale factor is applied. Due to differences in how sea level is calculated here and the calculations from the glacial isostatic adjustment (GIA) community stated above and additional allowance for the temporal scales modelled here, we discard those runs outside of 100 to 150 m sea level drop at LGM. In the early Pleistocene (older than 1 Ma) we take rough
sea level from Elderfield et al. (2012) and Rohling et al. (2014). Given the wide uncertainties in those records (the two records are occasionally anti-phase for example), we discard runs with an early Pleistocene sea level minimum outside of 50 and 120 m. The number of runs passing the sieve is not very sensitive to these bounds and changing them by 10 m only changes the number of runs passing the sieve by 1.

    Following the above glaciological controls, we apply the sedimentary constraints. The average removed bedrock thickness
of ≈ 70 m by Naylor et al. (2021) carries uncertainties which are difficult to assess as this is based on geomorphological interpretation of the mapped bedrock and reconstruction of the isostatically adjusted Pliocene landscape. Errors in the bedrock mapping depend on data coverage and any systematic biases thereof – this could introduce errors in bedrock elevation we estimate to be on the order of 10 m. Interpretation of this surface to produce a Pliocene bedrock surface may compound these errors five or ten fold. As such we assume an order of magnitude change for the lower bound of average bedrock removal – 7
m. The sieve is not sensitive to an upper bound beyond their estimate and so we take 70 m for this value.

    As the Alberta (Atkinson et al., 2020), Saskatchewan (Fenton et al., 1994), and northern US (**?**) data sets cover large areas in locations with excellent borehole coverage due to their natural resource industries, it is very unlikely that the volume of sediment in these areas could be off by more than a factor 2. We take half to twice these volumes as bounds. The St Lawrence



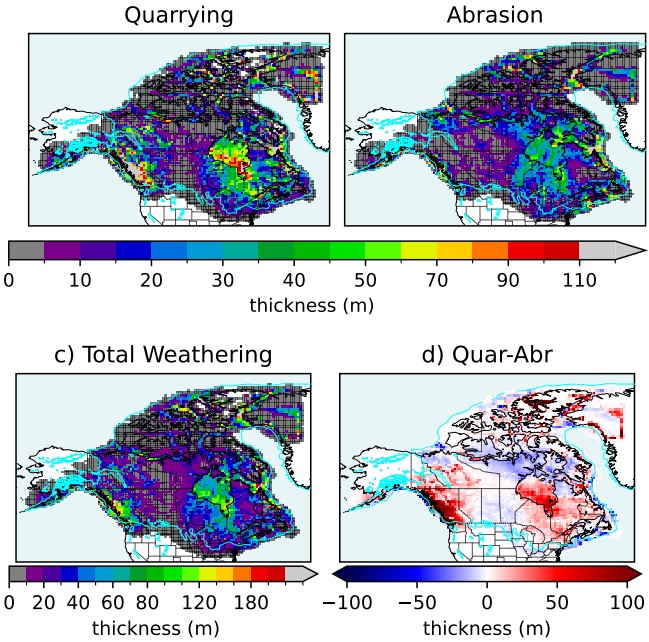

**Figure 9.** Mean depth of bedrock weathered over the Pleistocene due to a) quarrying and b) abrasion. Total weathering thickness is shown in c) and the difference between quarrying and abrasion is shown in d).

dataset of (Parent et al., 2021) covers a smaller region and so the comparison with the GSM scale is more tenuous. As such
we apply bounds of one third to three times this sediment volume. Finally, while the estimate of (Bell and Laine, 1985) covers
a large area like the first three datasets, it relies on seismic data which carriers uncertainties in the velocity models used to
convert travel time to depth, the interpretation, and the aquistion and processing steps. As such we widen these uncertainty
bounds further taken one quarter to four fold their sediment volume.

Applying all of these bounds in parallel to sieve the ensemble brings the number of runs from 728 to 16 (cf Fig. 8). The initial
regolith thicknesses for the latter surviving subset of runs spans 0 to 40 m, while no runs with 50 m initial sediment thickness
passed the sieving. This suggests that the mean Pliocene regolith was less than 50 m thick with no lower bound inferrable from
the above analysis.

## 6.2 Quarrying, Abrasion, and Overall Physical Weathering

Broadly, modelled quarrying and abrasion thickness shown in Fig. 9 is at similar scales and active in similar areas. Weathering
by both modes is high in the channels of the Canadian Arctic Archipelago and Hudson Strait and low on the islands. Since the
bedrock erodibility is constant, this illustrates the importance of topography and ice stream formation for weathering efficacy.
Quarrying is particularly effective in the Rockies where variable topography encourages streaming and melt-water formation



due to basal friction. Quarrying is also dominant in the Hudson Bay and Northwest Territories. Abrasion dominates in northeast mainland Nunavut, along the margin of the Quebec-Labrador sector, and other marine margins.

## 7 Discussion & Conclusions

It is thought that denudation of the Canadian shield occurred at some point following the intensification of Northern Hemispheric glaciation, removing a covering of regolith developed during the Pliocene or earlier (White, 1972; Clark and Pollard, 1998; Willeit et al., 2019). However, when sediment model parameters are tuned for agreement with the bedrock erosion depth estimate of Naylor et al. (2021), the sediment areal coverage over North America does not decrease much or in some cases actually increases. There are four potential reasons for this: 1. the sediment transport parameters should be tuned for a more transportive ice sheet, 2. the unstated uncertainties in bedrock erosion estimates of (Naylor et al., 2021) are large and lower values may be correct, 3. the missing fluvial, glacio-fluvial, and hillslope processes may be more important than assumed, and 4. there was little regolith cover in the Pliocene. The main sediment transport parameter is the sediment viscosity in the soft sediment deformation rheology relationship eq. **??**. While this parameter is largely unconstrained, decreasing the sediment viscosity results in peak sediment velocities which depart from observed sliding velocities in contemporary ice sheets. The uncertainties in the present day bedrock surface depth map of Naylor et al. (2021) are not stated and are likely far less than their Pliocene surface geomorphological reconstruction, the two inputs to their bedrock erosion estimate. It is difficult to assess how large these uncertainties are without an intimate understanding of the inputs and workflow. A detailed uncertainty assessment of these reconstructions would strongly facilitate interpretation and usability.

The fluvial role in North American sediment coverage following 2.58 Myr of Pleistocene glaciation remains to be quantified. Koppes and Montgomery (2009) point out that where sediment is mobilized by recent disturbance, a condition which a recent deglaciation likely satisfies, fluvial transport can increase "dramatically." Furthermore, the large amount of water availability during deglaciation and glacial lake drainage events may also increase this effect. Future work to encapsulate sediment transport due to glacial lake drainage events, glacio-fluvial transport, and interglacial fluvial and hill slope transport in a model similar to that presented here would help assess to what degree those processes can increase the amount of sediment transported during the Pleistocene. During interglacials chemical weathering could weaken bedrock and produce more quantities of regolith throughout the Pleistocene requiring even greater transport. Of course it is also possible that there was not enough Pliocene regolith to require an increased transport capacity to match the present day observation. If the change in North American hard bedded area across the Pleistocene was not significant, this would refute the regolith hypothesis for the mid-Pleistocene Transition.

Our understanding of the change in sediment coverage over the Pleistocene can be made more confident by quantifying the amount of Pleistocene aged sediment in significant depositional basins. Large scale geomorphological interpretation of large terrestrial deposits could help distinguish between the roles of glacial versus (glacio)fluvial transport. Notable terrestrial depositional basins for glacial sediment simulated by the model are: the area at the southern end of the glaciated Rocky and Coastal Mountains, southern margin tills in north-central USA, Western Canadian Sedimentary Basin, and the Hudson Bay





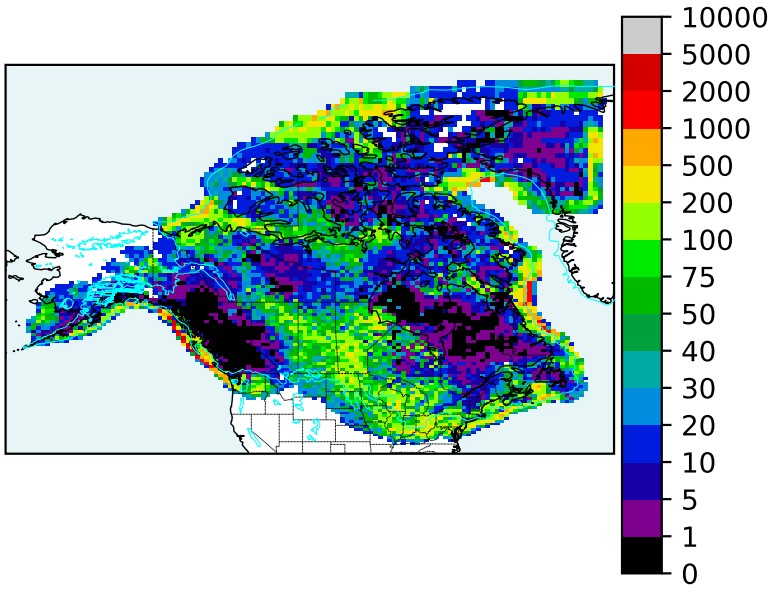

**Figure 10.** Mean ending sediment distribution across all parameter vectors and initial sediment thicknesses.

Lowlands. Pelletier et al. (2016) shows a thick sediment package in the Hudson Bay Lowlands whereas Taylor (2023) and (Shangguan et al., 2017) do not. We do not know of any reliable sediment volume constraint currently available in the Hudson Bay Lowlands.

The regional sediment volumes from the datasets of Fenton et al. (1994); Soller and Garrity (2018); Atkinson et al. (2020); Parent et al. (2021) provide important constraint where the present day sediment distributions of Pelletier et al. (2016); Shangguan et al. (2017); Taylor (2023) fall short. The sediment volume over southern Manitoba (Keller and Matile, 2021) is an outlier when considered alongside the other regional datasets. When applied in tandem with glaciological constraints as detailed in Sec. 6.1.1 these suggest a Pliocene regolith thickness of less than 50 m.

        Total sediment production is largely insensitive to the amount of starting sediment tested here (fig. 9, E1), though a small
trend does emerge when examining sediment production amounts as in fig. 7. Fig. 7 shows good agreement between the median value of Atlantic marine sediment in the model and the estimate of Bell and Laine (1985) at 20 m initial sediment thickness. Median modelled bedrock erosion in the reconstruction area of Naylor et al. (2021) lies below their estimate of ≈70 m for all initial regolith thicknesses. The Ehlers site erosion results are all in range of the wide bounds by Ehlers et al. (2006). The simulations with less starting sediment score better against the present day distribution of sediment. Additional estimates on
the spatial distribution of Pleistocene erosion depth at a catchment scale in more northern areas (e.g. Nunavut) would provide good constraint for the Pleistocene sediment budget and help improve understanding of glacial erosion processes. For example, estimates of erosion from terrestrial cosmogenic nuclide concentrations in till can be used to estimate mean catchment erosion rates at a scale comparable to ice sheet modelling (Staiger et al., 2006).



Quarried and abraded thicknesses are similar in magnitude and location with a few difference. Fig. 9 d) shows the quarrying-abrasion difference ($\int \dot{Q}dt - \int \dot{A}dt$). This highlights that quarrying is the dominant mode of physical weathering in the Cordillera, the Hudson Bay and parts of Quebec, and the fjords around Ellesmere island. Topographic troughs facilitate ice streaming and melt water production giving high subglacial cavity growth rates and thus quarrying. Also, variable topography results in higher basal strain heating and increased meltwater production, which explains the dominance of quarrying in the Cordillera.

The modelled weathered thickness in Greely Fjord ranges between 100-400 m where bathymetry ranges 200 at the head 500 at the mouth (all bathymetry numbers from Group, 2023). In Nansen Sound the modelled weathered thickness range from 250 to more than 400 m and the present day bathymetry is around 800 m deep. In the Laurentian channel area – which is between 80 and 450 m below its flanks – the model gives 100-140 m of weathering. As this feature is below the resolution of the model the local weathering could increase at higher resolution.

Interestingly the physically weathered thickness in the Hudson Bay matches the bathymetry there at present day. The weathered thickness ranges 80 to 200 m and the bathymetry range 60 to 250 m with more weathered areas correlating the deeper parts of the bay. This supports the hypothesis that prior to glaciation the Hudson Bay region was sub-aerial and perhaps a large river system which was subsequently excavated during glaciation (the Bell River Hypothesis, Corradino et al., 2021; Sears and Beranek, 2022). Connecting these modelled weathering rates with topographic lows presupposes increased pre-glacial erodibility in these areas as the present day topography was used to simulate the weathering and topographic lows beget topographic lows. Nevertheless, for the Hudson Bay at least, it is a priori plausible that this area was more erodible as it is underlain by softer carbonates (Hanna et al., 2018). These differences in bedrock erodibility may be seen as physiographic character which encourages bedrock weathering or as material weakness within the rock due to lithology, fracture, or jointing (e.g. Hooyer et al., 2012).

In this model, most of the sediment is transported to marine areas (see fig. 10) An exploration program (or re-analysis of past programs) over a few offshore basins would provide good constraint to the Quaternary sediment budget. Perhaps the most significant and accessible location for sediment budget constraint is the Saglek Basin area at the mouth of Hudson Strait. The Saglek basin is distal from large present day fluvial systems and its Pleistocene sediment volume is probably dominated by ice stream transport. This area has been of interest for oil & gas exploration in the past and there are several wells and 2D seismic lines in the area (Jauer and Budkewitsch, 2010). This basin would also be the deltaic depocenter for the hypothesized Bell River which is supposed to have carried sediment from large parts of North America across a terrestrial Hudson Bay prior to glaciation. This far-flung dispersion is support by recent detrital zircon work showing some of the pre-glacial sediment in the Saglek basin came from the Colorado Plateau (Sears and Beranek, 2022). Similarly, the Laurentian Channel and Scotian shelf have been targeted for exploration with existing data coverage. Additional areas include the north end of Baffin Bay and Canada Basin – though exploration in Canada Basin in particular has proved especially difficult (Stashin, 2021).

Based on the discussion above, we would single out two desired model inputs to make the basal processes representations more realistic. Rougher terrains will have more bedrock rising above the unconsolidated sediment fill – the hard bedded area calculation should account not only for the mean thickness of sediment but for topographic roughness as well. High resolution



(2 m) digital elevation models (DEMs) are available which would allow roughness calculations at the relevant several metre
520   scales (Porter et al., 2018). This would also allow incorporating basal roughness in the sediment and subglacial hydrology
models as a field from DEMs than a spatially uniform parameter. It is not clear, however, what measure of basal roughness
(e.g. standard deviation, slope, power spectra, etc.) would be most applicable. Bedrock erodibility likely varies spatially as well,
whereas in the model it is currently treated as a uniform uncertainty parameter. Quarrying has been shown to be controlled by
jointing/fracture spacing (Hooyer et al., 2012) and abrasion by rock hardness (Engelder and Scholz, 1976). Incorporating maps
525   of erodibility in the respective modes based off lithology and other attributes would be a major improvement.





*Code and data availability.* Data sets and code will be made available upon final acceptance for publication. We plan on providing sediment cover and bed elevation anomaly during interglacials for a range of parameterizations and welcome reasonable suggestions for additional outputs.

## Appendix A:  Present Day Sediment Estimates

| Unit | Description | Thickness (m) |
|---|---|---|
| I | Glacier ice | not used |
| O | Organic > 2 m | 3 |
| E | Eolian | 3 |
| Cv | Colluvial veneer | 1 |
| C | Colluvial seds, undif | 4 |
| A | Alluvial seds, undif | 4 |
| Ln | Lacustrine (littoral, nearshore)*below modern lakes | 5 |
| Lo | Lacustrine (offshore)*below modern lakes | 10 |
| Mn | Marine (littoral, nearshore)*below modern seas | 5 |
| Mo | Marine (offshore)*below modern seas | 10 |
| GMn | Glaciomarine and marine (littoral, nearshore) | 5 |
| GMv | Glaciomarine and marine (veneer) | 1 |
| GMo | Glaciomarine and marine (offshore) | 10 |
| GLn | Glaciolacustrine and lacustrine (littoral, nearshore) | 5 |
| GLo | Glaciolacustrine and lacustrine (offshore) | 10 |
| GFp | Glaciofluvial seds (outwash plain) | 10 |
| GFc | Glaciofluvial seds (ice-contact) | 25 |
| Tv | Glacial seds (till veneer) | 1 |
| Tb | Glacial seds (till blanket) | 7 |
| Th | Glacial seds (hummocky till) | 10 |
| Tm | Glacial seds (moraine complex) | 35 |
| Wv | Weathered bedrock (regolith veneer) | 1 |
| W | Weathered bedrock (regolith, undiff) | 3 |
| V | Bedrock (Quaternary volcanic rocks and deposits) | 0 |
| R | Bedrock, undiff ( > 75% outcrop) | 0 |

**Table A1.** Assumed sediment thicknesses by surficial geologic unit used by Taylor (2023) for reconstructing the present day sediment thickness over North America were direct observations were lacking (outside the study areas of Soller and Garrity (2018); Parent et al. (2021); Russell et al. (2017); Smith and Lesk-Winfield (2010)).





## Appendix B: Description

The sediment model applied in the GSM is designed after that of Hildes (2001). Both employ separate descriptions of abrasion and quarrying for the erosive part, although Melanson (2012) also provides an optional empirical erosion as well as a choice between Boulton and Hallet style erosion whereas Hildes (2001) uses only Hallet. Hildes (2001) input a map of substrate lithology and track the distribution of clast lithology within a given cell, calculating the relative hardness of substrate and abrasive. Abrasion is switched off when clast hardness is lower than the substrate, whereas Melanson (2012) neglects the clast-substrate hardness differential. Hildes (2001) also uses lithology dependent quarrying rate factors, although the accuracy of this is questionable as quarrying primarily relies on the density of pre-existing joints and weaknesses in the substrate (Hooyer et al., 2012; Iverson, 2012). Joint and fracture density is not easily quantified at the relevant scales and is highly variable within the dimension of a grid cell (e.g. km to 100's km). Hildes et al. (2004) and Melanson (2012) both use estimates of cavitation (from basal water pressure) in the lee side of basal protrusions to estimate stress regimes and thus quarrying rates. Both models give englacial transport rates through advection, however, Hildes (2001) omits subglacial transport.

### B1   Abrasion Law

Abrasion occurs from contact between two surfaces either through polishing or through striation (Benn and Evans, 2010). Polishing is the process by which the abrading surface removes protrusions in the bed reducing its roughness. Striation is the process by which stress caused by asperities in the abrading surface generate fracture sets which accumulate as linear troughs behind the asperity. Abrasion in the GSM is given by (Melanson et al., 2013):

$$\dot{A} = \exp\left(-h_{sed}/\tilde{h}_{sed}^{\star}\right)\frac{k_{abr}}{HV^{\star}}\sum_{R}C_b\left(R\right)F_N\left(R\right)\left|v_{par}\left(R\right)\right| \tag{B1}$$

The exponential represents a shielding coefficient – decreased contact between the abrading surface and the substrate as sediment accumulates. $h_{sed}$ is the sediment layer thickness and $\tilde{h}_{sed}^{\star}$ is a characteristic thickness for shielding (shielding factor). This shielding factor is taken by Hildes (2001) to represent the areal fraction of the rough bed within a given cell which is covered by sediment. Hildes (2001) calculates this as

$$\tilde{h}_{sed}^{\star} = \tilde{d}_1\theta_{mean} + \tilde{d}_0 \tag{B2}$$

where $\tilde{d}_1 = 10.0$m and $\tilde{d}_0 = 2.0$m and bed dip $\theta_{mean} \in [0, 0.45]$ (deg) so that $\tilde{h}_{sed}^{\star} \epsilon [2.0, 6.5]$. Here, since this range of bed dips was defined using coarser resolution digital elevation maps (30-1000 m horizontal resolution, § 2.1 in Hildes, 2001), the assumed a priori range of $\tilde{h}_{sed}^{\star}$ is widened to $[2.0, 20.0]$ as higher resolution could pick up steeper (rougher) beds. The abrasion coefficient $C_{abr}^{\star}$ is a tuning parameter defined as

$$C_{abr}^{\star} = k_{abr}/HV^{\star}. \tag{B3}$$

The abrasion wear coefficient, $k_{abr}$, is shown in Hildes (2001) to vary over 2 orders of magnitude from $[0.002, 0.3]$ on the basis of reference to the literature. $HV^{\star}$ is the Vickers hardness. Since the substrate hardness does not appear here it is



implicitly assumed that the substrate is softer than the abrading material. From review of the literature, Hildes (2001) gives a range of Vickers hardness values of $HV^\star \in \left[0.3 \times 10^{-9}, 12 \times 10^{-9}\right]$ Pa. This gives the range of abrasion coefficient: $C_{abr} \in \left[1.7 \times 10^5, 1.0 \times 10^9\right]$. Experiments by Iverson (1990) (granitic fragments frozen into ice and pressed into a sliding marble substrate) found that abrasion in their experiments more closely resembled the abrasion law of Hallet (1979) than that of Boulton (1979).

$C_b$ is entrained debris concentration at a range of sediment radii ($R$), calculated from the englacial sediment concentration at the bottom (first) layer, $C_{eng}^1$ (see § **??**) assuming volume concentration is equal to areal concentration:

$$C_b(R) = \frac{C_{eng}^1 \exp\left\{-\left(\log_{10}(R) - \log_{10}(\mathrm{E}[R])\right)^2 / (2\log_{10}\mathrm{Var}[R])\right\}}{\pi R^2 \sum \exp\left\{-\left(\log_{10}(R) - \log_{10}(\mathrm{E}[R])\right)^2 / (2\log_{10}\mathrm{Var}[R])\right\}} \tag{B4}$$

$F_N$ is the normal force between bed and abrading particle, $v_{par}$ is the speed of the abrasive across the bed. These last two factors are calculated according to two different models of subglacial abrasion: the Boulton model assumes the normal force is
proportional to the effective pressure $N_{\mathrm{eff}}$ while particle velocity is reduced by a particle size, effective pressure, and ice flow dependent drag terms.

$$F_N = A_e P_e \tag{B5}$$

$$v_{par}(R) = |v_s| - 2B_g^\star R\left(\frac{\mu^\star P_e A_e}{A_r}\right)^3 - \frac{P_m P_e A_e}{2L_{fus}\rho_i A_r R} \tag{B6}$$

Conversely, the Hallet model assumes the normal force is proportional to particle size and normal velocity (from normal component ice velocity due to subgrid bed undulation and basal melt rate). Hallet's particle velocity also assumes a slow down from bed contact, assuming equilibrium between friction with the bed and "drag" from the ice.

$$F_N(R) = f_{bed}^N \frac{4\pi\eta R^3}{\tilde{R}^2 + R^2} v_n \tag{B7}$$

$$v_{par} = |v_s|\left(1 - \frac{\mu^\star f_{bed}^N \sin\theta}{f_{bed}^T}\right) - \frac{\mu^\star f_{bed}^N}{f_{bed}^T}\dot{b}_{melt} \tag{B8}$$

Abrasion is turned off when abrading particle velocity reaches zero, ice becomes cold-bedded, or $N_{\mathrm{eff}}$ reaches zero (flotation).

## B2  Quarrying Law

Quarrying is also an erosive process like abrasion, but produces larger ($> 1\,\mathrm{cm}$) fragments of rock, taking advantage of joints
(weaknesses in the rock Benn and Evans, 2010) as seen in the control of quarried surface orientation toward pre-existing joint and fracture sets over sliding direction (Hooyer et al., 2012). Stress gradients due to cavity growth concentrate force on bedrock



steps where these forces can exceed 50% higher than ambient ice-bed pressures (Iverson, 1991). Separation pressure, $P_s$, is given in Melanson et al. (2013); Cuffey and Paterson (2010) as:

$$P_s = P_{ice} - \frac{\tau_b}{\pi \zeta^\star} \tag{B9}$$

where $P_{ice}$ is ice overburden, $\tau_b$ is basal shear stress (assumed to be driving stress), and $\zeta^\star$ is basal roughness (protrusion height over distance between).

$\zeta^\star$ is similar to the basal roughness $h_r$ in the linked-cavity hydrology formulation. Additionally, Melanson (2012) assumes the grid-scale cavity extent (important for quarrying rate) is proportional to the residual pressure (difference between water pressure and separation pressure). However cavity extent is given in linked-cavity hydrology directly by the grid-scale sub-

glacial water thickness $h_{wb}$. Because of this dependence on cavity formation, where quarrying rates are large, linked-cavity drainage may be dominant.

The quarrying rate in Melanson et al. (2013) is given by:

$$\dot{Q} = \exp\left(-h_{sed}/\tilde{h}_{sed}^\star\right) C_{quar}^\star \left(\frac{P_w - P_s}{\tilde{P}_r}\right)^{n_p^\star} \tag{B10}$$

with $C_{quar}^\star$ the quarrying coefficient, $n_p^\star$ an exponent controlling residual pressure contribution, $\tilde{P}_r$ the characteristic residual

pressure ($\approx 18 kPa$)

**B3  Empirical Erosion Law**

The empirical erosion rate used in the GSM is (Melanson et al., 2013)

$$\dot{E}_{emp} = \exp\left(-h_{sed}/\tilde{h}_{sed}^\star\right) C_{emp}^\star |\tau_b||v_s| \tag{B11}$$

with $C_{emp}^\star$ a coefficient and $v_s$ the sliding velocity.

**B4  Englacial Transport: Entrainment, Mixing, and Advection**

Englacial transport in the GSM is (Melanson et al., 2013)

$$\frac{\partial C}{\partial t} = -\nabla \cdot C\boldsymbol{v}_i - \frac{\partial(C\dot{V}_{net})}{\partial z} + V_{mix} \tag{B12}$$

The lateral velocities $v_i$ are a function of lateral and vertical position. The entrainment/deposition rate $V_{net}$ is a function of lateral position. The vertical mixing rate $V_{mix}$ is diffusion of sediment concentration between layers which varies laterally and

vertically. Within the GSM, vertical diffusion and lateral advection are treated separately with lateral advection calculated first for a given time step. A tri-diagonal matrix solver is used to calculate the sediment concentrations due to vertical diffusion. Vertical mixing is given by:

$$V_{mix} = \frac{\partial}{\partial z}\left(D\frac{\partial C}{\partial z}\right) \tag{B13}$$



$$D = \tilde{D}^\star \left( \frac{|v_s|}{\tilde{v}_s} \right) e^{-z/\tilde{z}^\star} e^{-h_{sed}/\tilde{h}^\star} \tag{B14}$$

Where $\tilde{D}^\star$, $\tilde{z}^\star$, and $\tilde{h}^\star$ are scale factor parameters and $\tilde{v}_s$ is a scale factor based on typical sliding velocities. Vertical diffusion decreases with increased sediment cover because where there is more soft sediment there will be less folding within the basal ice and thus less vertical (eddy) diffusion. This is because basal stress is balanced by viscous or plastic forces within the soft-deformation layer instead of by the process of folding in basal ice.

Sediment is entrained within the ice by regelation (Iverson and Semmens, 1995). The rate of regelation of ice into the sediment is given by:

$$v_r = K_s \frac{N_{eff}}{l_a} \tag{B15}$$

where $K_s$ is the conductivity of ice into the sediment layer (defined in Hildes (2001, eqn. 4.4 and 4.5)) and $l_a$ is the englacial sediment array thickness. This array thickness acts to limit sediment entrainment as the medium transitions from ice supported

to clast supported and the medium can no longer regelate around sediment grains. The effective array depth is the series of smoothed step functions of sediment thickness within each basal grid layer:

$$l_a = \sum_{k_z} l_{mod}(k_z) \Delta z(k_z) \tag{B16}$$

$$l_{mod}(z) = 0.5 \{ \tanh[20(C(z) - C^\star_{crit})] + 1 \} \tag{B17}$$

$V_{net}$ term calculated as the net of entrainment (due to regelation) and deposition (due to basal melt), giving:

$$V_{net} = v_r - \frac{\dot{b}_{melt} C(z=0)}{1 - \phi^\star} \tag{B18}$$

where the sediment concentration in the bottom layer is deposited with porosity $\phi^\star$ when melting ($\dot{b}_{melt}$) occurs. This implies recently entrained sediment has concentration $C(z=0) = 1 - \phi^\star$. Entrainment rate is capped at value $v^\star_{max}$ and layer concentration is capped at $C^\star_{max}$.

Discretization above the ice-sediment interface used for modelling entrainment, mixing, and advection of sediment in the ice uses an exponential grid. This exponential grid concentrates resolution in the most dynamic basal layers where mixing/entrainment velocities are highest. The ice-sediment interface moves up as sediment is deposited and down as bedrock is eroded or denuded. The grid is chosen uniform in $\xi$ and transformed using:

$$z_{ice} = \frac{z_{max}}{\sum [e^{\xi/z_0}]} e^{\xi/z_0} \tag{B19}$$

where $z_0 = 15$ is the exponential grid spacing parameter and $z_{max} = 23$ is the full height of the entrainment grid in the ice. This grid is depicted in fig. B1.



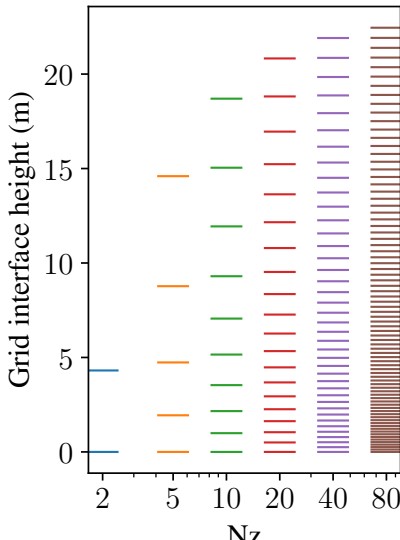

**Figure B1.** Depiction of the vertical englacial sediment grid for various resolutions.

## B5   Subglacial Transport Law

Basal sediment transport is governed by the sediment continuity relationship given in Melanson et al. (2013):

$$\frac{\partial h_{sed}}{\partial t} = -\nabla \cdot \boldsymbol{Q}_s + \dot{E} - V_{net} \tag{B20}$$

where sediment thickness through time is given as the sum of the convergence of subglacial transport through deformation, erosion rate $\dot{E}$ (empirical or process), and entrainment rate. This acts with englacial transport in eqn. B12 to give the total sediment transport and production.

Subglacial transport from soft sediment deformation occurs as basal shear stress from the overbearing ice sheet is applied to a layer of unconsolidated till assumed to behave as a Coulomb plastic in the manner of Iverson (1985). The constitutive
relationship presented therein is between the deformation rate tensor (**D**) and the stress deviator tensor (**T**):

$$\mathbf{D} = \begin{bmatrix} \frac{\partial v_x}{\partial x} & \frac{1}{2}\left(\frac{\partial v_x}{\partial y} + \frac{\partial v_y}{\partial x}\right) & \frac{1}{2}\left(\frac{\partial v_x}{\partial z} + \frac{\partial v_z}{\partial x}\right) \\ \frac{1}{2}\left(\frac{\partial v_y}{\partial x} + \frac{\partial v_x}{\partial y}\right) & \frac{\partial v_y}{\partial y} & \frac{1}{2}\left(\frac{\partial v_y}{\partial z} + \frac{\partial v_z}{\partial y}\right) \\ \frac{1}{2}\left(\frac{\partial v_z}{\partial x} + \frac{\partial v_x}{\partial z}\right) & \frac{1}{2}\left(\frac{\partial v_z}{\partial y} + \frac{\partial v_y}{\partial z}\right) & \frac{\partial v_z}{\partial z} \end{bmatrix} \tag{B21a}$$

$$\mathbf{T} = \begin{bmatrix} \tau_{xx} + p & \tau_{xy} & \tau_{xz} \\ \tau_{yx} & \tau_{yy} + p & \tau_{yz} \\ \tau_{zx} & \tau_{zy} & \tau_{zz} + p \end{bmatrix} \tag{B21b}$$





where $v_i$ is the $i^{\text{th}}$ component of velocity, $\tau_{ij}$ are the components of deviator stress, and $p = -\frac{1}{3}\left(\tau_{xx} + \tau_{yy} + \tau_{zz}\right)$. Iverson (1985) then uses a generalization of the Coulomb criterion for plastic yield, the extended Von Mises (Von Mises, 1913) criterion — Drucker and Prager (1952, eqn. 2):

$$II_T^{1/2} = k + \alpha p \tag{B22}$$

where $II_T$ is the second invariant of the stress deviator tensor, $k$ is the cohesion and $\alpha$ is the tangent of the angle of internal friction. This is

$$II_T = \frac{1}{2}\left[\operatorname{tr} T^2 - \operatorname{tr} T^2\right] \tag{B23}$$

$$= \frac{1}{2}\left[(\tau_{xx} + p)^2 + (\tau_{yy} + p)^2 + (\tau_{zz} + p)^2 + 2\left(\tau_{xy}^2 + \tau_{yz}^2 + \tau_{zx}^2\right)\right]. \tag{B24}$$

Iverson (1985) then works out his constitutive equation:

$$\mathbf{T} = 2\mu\mathbf{D} + \frac{k + \alpha p}{II_D^{1/2}}\mathbf{D} \tag{B25}$$

where

$$II_D^{1/2} = \begin{cases} \frac{1}{2\mu}\left[II_T^{1/2} - (k + \alpha p)\right] & II_T^{1/2} > k + \alpha p \\ 0 & II_T^{1/2} \leq k + \alpha p \end{cases}. \tag{B26}$$

In the context of the shallow ice approximation (no lateral or longitudinal shear components) and assuming no vertical transport (vertical length scales are much shorter than lateral when comparing the horizontal dimensions of ice sheets against vertical dimensions of till beds), the deformation and stress tensors become:

$$\mathbf{D} = \begin{bmatrix} 0 & 0 & \frac{1}{2}\left(\frac{\partial v_x}{\partial z} +\right) \\ 0 & 0 & \frac{1}{2}\left(\frac{\partial v_y}{\partial z}\right) \\ \frac{1}{2}\left(\frac{\partial v_x}{\partial z}\right) & \frac{1}{2}\left(\frac{\partial v_y}{\partial z}\right) & 0 \end{bmatrix} \tag{B27a}$$

$$\mathbf{T} = \begin{bmatrix} 0 & 0 & \tau_{xz} \\ 0 & 0 & \tau_{yz} \\ \tau_{zx} & \tau_{zy} & 0 \end{bmatrix} \tag{B27b}$$

, this also assumes incompressible shear such that $\tau_{xx} = \tau_{yy} = \tau_{zz} = -p$. This $p$ is the effective pressure with sediment overburden, $N_{eff} + \Delta\rho gz$. Using a similar model of subglacial deformation, Pollard and Deconto (2009) assumes $N_{eff} = 0$ so that the normal stress is only applied by the sediment overburden. The model in the GSM follows Jenson et al. (1995) and take the normal stress as $N_{eff} + (\rho_s - \rho_w)gz$ with $z$ the depth into sediment. Jenson et al. (1995) then shows that the independent lateral components are:

$$\tau_{xz} = (2D_0)^{(n_s - 1)/n_s}\mu_0\left(\frac{dv_x}{dz}\right)^{1/n_s} + c + (N_{eff} + \Delta\rho gz)\tan\phi^\star \tag{B28a}$$





$$\tau_{yz} = (2D_0)^{(n_s-1)/n_s} \mu_0 \left( \frac{dv_y}{dz} \right)^{1/n_s} + c + (N_{eff} + \Delta\rho gz)\tan\phi^\star \tag{B28b}$$

$\tau_{xz}$ and $\tau_{yz}$ are the x/y components of ice basal shear stress. Rearranging for $v_i$ ($i^{\text{th}}$ component of sediment velocity) gives:

$$v_i = \int \left[ \frac{\tau_{iz} - c - (N_{eff} + \Delta\rho gz)\tan\phi^\star}{(2D_0)^{\frac{n_s-1}{n_s}} \mu_0} \right]^{n_s} dz \tag{B29}$$

$$= \frac{[\tau_{iz} - (c + N_{eff}\tan\phi^\star + \Delta\rho gz\tan\phi^\star)]^{n_s+1} / (n_s+1)}{\left[ (2D_0)^{\frac{n_s-1}{n_s}} \mu_0 \right]^{n_s} \Delta\rho g\tan\phi^\star} + \gamma_1 \tag{B30}$$

Where $\gamma_1$ is the constant of integration. Assuming a reference frame of zero background flow of sediment, this constant is $\gamma_1 = 0$. At the ice-sediment interface, $z = 0$,

$$v_i = \frac{[\tau_{iz} - c - N_{eff}\tan\phi^\star]^{n_s+1}}{(n_s+1)(2D_0)^{n_s-1}\mu_0^{n_s}\Delta\rho g\tan\phi^\star} \tag{B31}$$

Deformation thickness is found by solving for $v_i(z = z_d) = 0$:

$$0 = \tau_{iz} - c - N_{eff}\tan\phi^\star - \Delta\rho gz_d\tan\phi^\star \tag{B32}$$

$$\Rightarrow z_d = \frac{\tau_{iz} - c - N_{eff}\tan\phi^\star}{\Delta\rho g\tan\phi^\star} \tag{B33}$$

Integrating once more to get flux over the deformation thickness,

$$Q_i = \int_{z_d}^{0} v_i dz = -\int_{z_d}^{0} \frac{[\tau_{iz} - c - N_{eff}\tan\phi^\star - \Delta\rho gz\tan\phi^\star]^{n_s+1}}{\left[ (2D_0)^{\frac{n_s-1}{n_s}} \mu_0 \right]^{n_s} \Delta\rho g\tan\phi^\star (n_s+1)} dz \tag{B34}$$

$$= \frac{(\Delta\rho g\tan\phi^\star)^{n_s}()^{n_s+1}}{\left[ (2D_0)^{\frac{n_s-1}{n_s}} \mu_0 \right]^{n_s} (\Delta\rho g\tan\phi^\star)^2 (n_s+1)(n_s+2)} \tag{B35}$$

$$Q_{sx} = \begin{cases} 0 & \text{ice floating}||\text{freezing}||H \le 0. \end{cases}$$

$$Q_s = \begin{cases} h_{sed} \le z_p & \frac{\left[ \frac{-(\tau - c_{cohe} - N_{eff} + (\rho_s - \rho_w)g\tan\phi h_{sed})^{n_s+2}}{(n_s+2)(\rho_s - \rho_w)g\tan\phi} - (\tau - c_{cohe} - N_{eff}\tan\phi)^{n_s+2} - [\tau - c_{cohe} - (N_{eff} + (\rho_s - \rho_w)gh_{sed})\tan\phi]^{n_s+1}h_{sed} \right]}{(n_s+1)(2D_0)^{n_s-1} * \mu_0^{n_s}(\rho_s - \rho_w)g*\tan\phi} \\ h_{sed} > z_p & \frac{-1}{(n_s+1)(n_s+2)(2D_0)^{n_s-1}\mu_0^{n_s}[(\rho_s - \rho_w)g\tan\phi]^2} \left[ (-\tau - c_{cohe} - N_{eff}\tan\phi)^{n_s+2} \right] \end{cases}$$

## 695 Appendix C: Model Verification

We verify the model in three ways: symmetry of solutions, convergence under increasing resolution, and mass conservation. All model solutions were found to be symmetric. The temporal convergence tests are in § C1, spatial in § C2, number of bed dips in the abrasion calculation in § C3, number of grain sizes in the abrasion calculation in § C4, and number of vertical levels in the englacial transport grid in § C5. Mass conservation test is in § C6.





**Figure B2.** Flow chart of sediment model algorithm execution



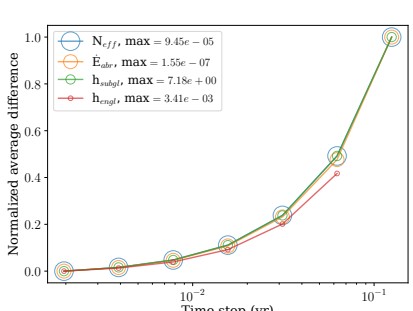

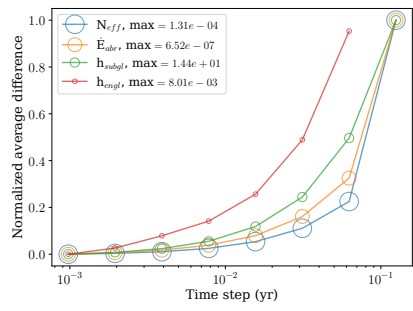

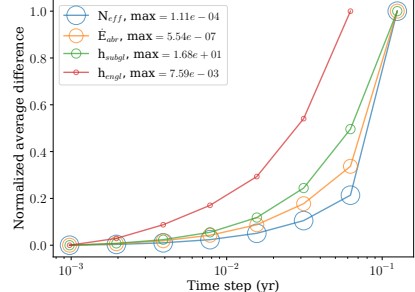

(a) Steady state, constant melt, 2.5 mm/yr

(b) Sinusoidal melt, 50 year period, 0 to 2.5 mm/yr

(c) Same as C1b plus basal sliding feedback and 0 to 3.5 mm/yr

**Figure C1.** Convergence with decreasing time step. Each metric is normalized such that the scale is consistent across metrics. The normalization factor shown in the legend (max).

## C1 Temporal convergence test

As a test of convergence under increasing temporal resolution (decreasing time step length) the sediment model was run to steady state under SHMIP scenario A (constant 2.5 mm/a de Fleurian et al., 2018), a sinusoidal melt forcing with constant sliding velocity, and sinusoidal melt forcing with coupled basal velocity. The convergence is shown in fig. C1. Under this scenario quarrying never switched on owing to the low driving stress of the setup, but the quarrying rate is numerically similar to (and simpler than) abrasion. Convergence is shown for the effective pressure $N_{eff}$, abrasion rate $\dot{E}_{abr}$, subglacial sediment thickness $h_{subgl}$, and englacial sediment thickness $h_{engl}$. The effective pressure solution converged most quickly followed by abrasion rate, and lastly the subglacial and englacial convergence were about the same (englacial not captured for highest time step).

## C2 Spatial convergence test

Here we show the effect of varying spatial resolution on the model solution. The models are run to steady state with prescribed melt and basal velocity (1.75 m/a ice and 2.0 m/a respectively) in the SHMIP setup (de Fleurian et al., 2018). The square root ice sheet flowline length from divide to toe is 100 km and the number of cells varied between 2,45. The model solution at each resolution is linearly interpolated to the highest resolution grid and the absolute sum of the difference against the highest resolution solution is used for the error.

Fig. C2 shows the convergence of model solutions (same set as fig. C1) at increasing spatial resolution (shorter cell width).

## C3 Number of bed dips convergence test

Here we show convergence under an increasing number of bed dips relevant for calculating normal force and velocities in the abrasion rate calculation. The abrasion rate model solution converges with an increasing number of dips. To optimize model performance, the number of bed dips was chosen at NUMBEDDIP = 10 (see fig. C3). Resolutions tested are $\{5i\}$ for $i \in [1, 18]$, $i \in \mathbb{N}$ for uniform distribution of angles, $\theta_{dip} \epsilon [-34°, +34°]$.

## C4 Number of grains size bins convergence test

The grain size distribution is used to calculate normal force and particle velocity in the abrasion rate. The abrasion rate solution converges with an increasing number of bins in the grains size distribution (fig. C4). Between 5 and 360 bins were tested and showed small error between the lowest and highest number of bins. As such 10 bins was chosen to optimize model performance.

## C5 Englacial grid resolution test

The englacial grid is used in the vertical diffusion of sediment during englacial transport. A depiction of the non-linear grid is shown in fig. C5. The number of englacial grid levels was tested between 2 and 80, with 10 chosen to optimize model performance (10 bins had $< 10\%$ error relative to highest bin count, see fig. C5).





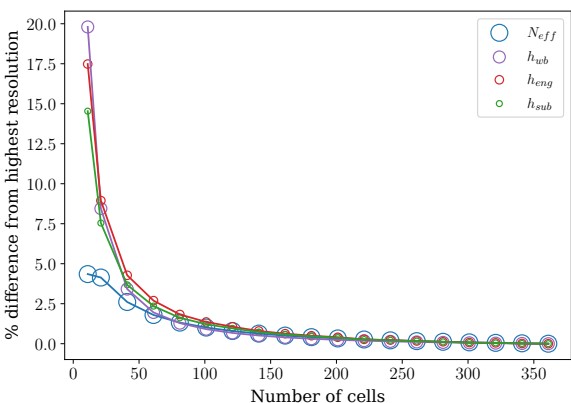

**Figure C2.** Difference in mean flowline solutions for unsteady SHMIP square root ice sheet topography as a function of increasing spatial resolution, at the end of 10 kyr run.

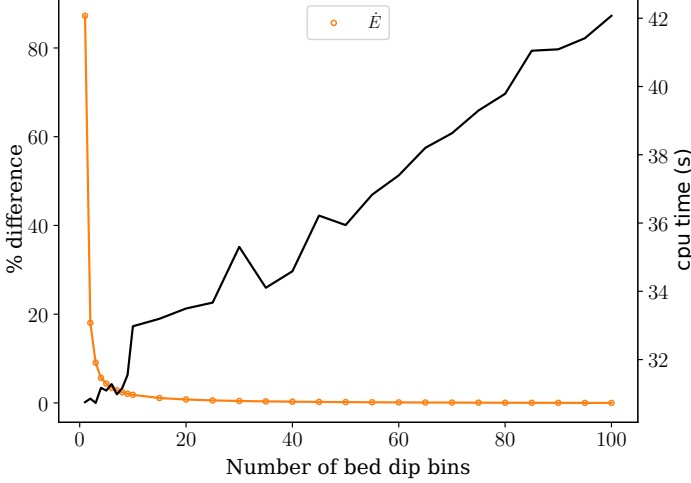

**Figure C3.** L2 norm between each resolution and the highest resolution run, normalized by highest resolution run.





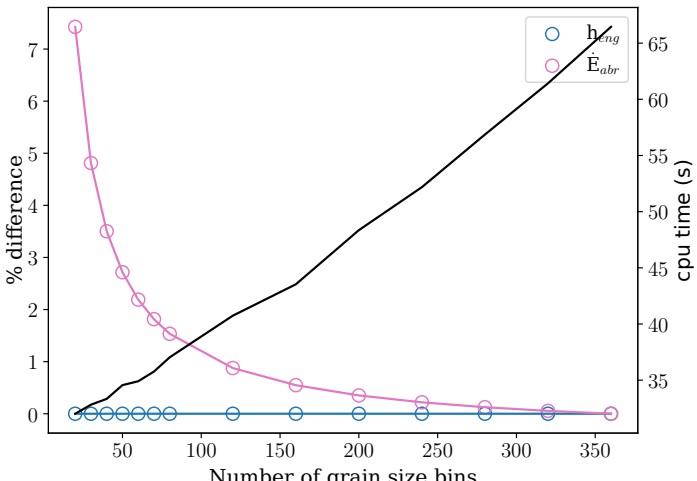

**Figure C4.** L2 norm between each resolution and the highest resolution run, normalized by highest resolution run.

## C6 Mass conservation test

The integral of the total erosion and removal of sediment over the ice sheet extent less the subglacial and englacial fluxes at the margin will give the total sediment within the ice sheet extent (subglacial and englacial). To test mass balance with unsteady input, we applied a sinusoidal meltwater forcing (eqn. **??**) to the $SQRT\_TOPO$ setup with a basal sliding velocity coupled to effective pressure. The sediment mass balance is calculated as the balance of inputs and outputs:

$$net_{sed}^{t_i} = \int_0^{t_i} \left( \int_{\mathcal{A}} \dot{E}_{abr} + \dot{E}_{quar} da - \oint_{\mathcal{S}} \left[ \mathbf{Q}_{subgl} + \mathbf{U}_b \int_0^{z_{engmax}} C_{eng} dz \right] \cdot \mathbf{n} dS \right) d\tau - \mathrm{h}_{subgl}^{initial} \tag{C1}$$

$$ERR_{sed}^{t_i} = \frac{\left| net_{sed}^{t_i} - V_{sed}^{t_i} \right|}{V_{sed}^{t_i}} \tag{C2}$$

where $V_{sed}^{t_i}$ is the total subglacial and englacial sediment within the ice sheet extent at time step $t_i$.

## Appendix D: Sediment Loading and Isostatic Adjustment Test

We initialize the bed with the present day isostatic equilibrium, impose the sediment load (increasing from experiment to experiment) and run to near equilibrium over 40 kyr. For these tests the $CO_2$ was set to 700 ppm to suppress ice growth (no ice developed), this melted ice over Ellesmere and the stub Greenlandic ice sheet. The change in topographic elevation (less sediment thickness) is plotted in fig. D1 and shows increased subsidence with increasing sediment load.

## Appendix E: Produced and pre-existing sediment

## Appendix F: Bed Dip Distribution for Abrasion Calculation

In the case of the Hallet abrasion model, the weathering rate is dependent on the angle of ice flow with respect to the bed. In order to account for the subgrid distribution of bed dip angle, we use the 2m lateral resolution ArcticDEM digital surface model (DSM) (Porter et al., 2023).

A given point and those above and to the left of that point all lie within a plane and give two vectors lying within that plane.

$$\boldsymbol{P1P0} = \boldsymbol{P1} - \boldsymbol{P0} \tag{F1}$$



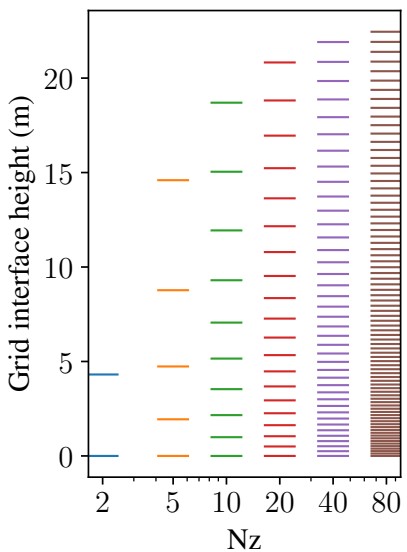

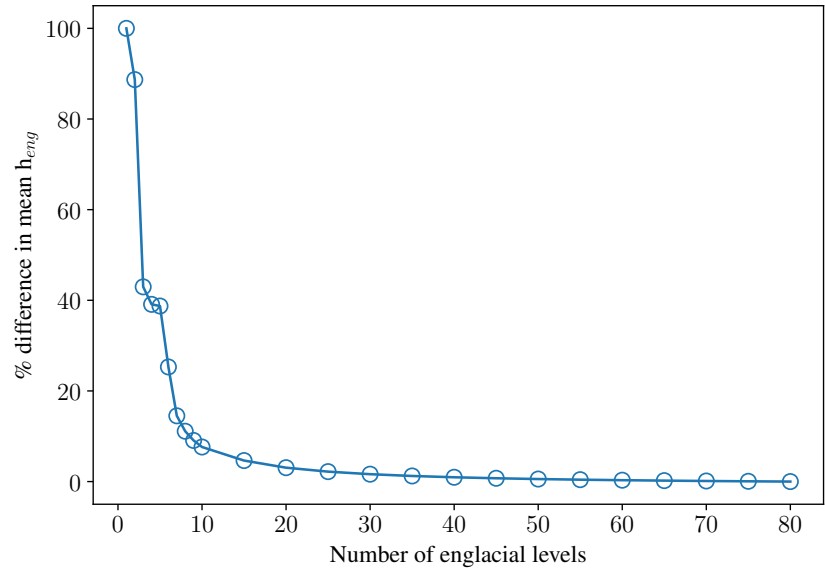

**Figure C5.** Convergence of englacial sediment thickness with more levels in the exponential englacial sediment grid.





**Figure C6.** Mass conservation for sinusoidal melt input Mass balance for subglacial hydrology and basal sediment models given square root ice sheet topography and sinusoidal ice sheet basal mass balance (ice thickness, $m/a$), $Gb = 3.5 \times 10^{-3}/2 \sin 2\pi t/1000 + 3.5 \times^{-3}/2$. Basal sliding velocity calculated from driving stress and effective pressure.




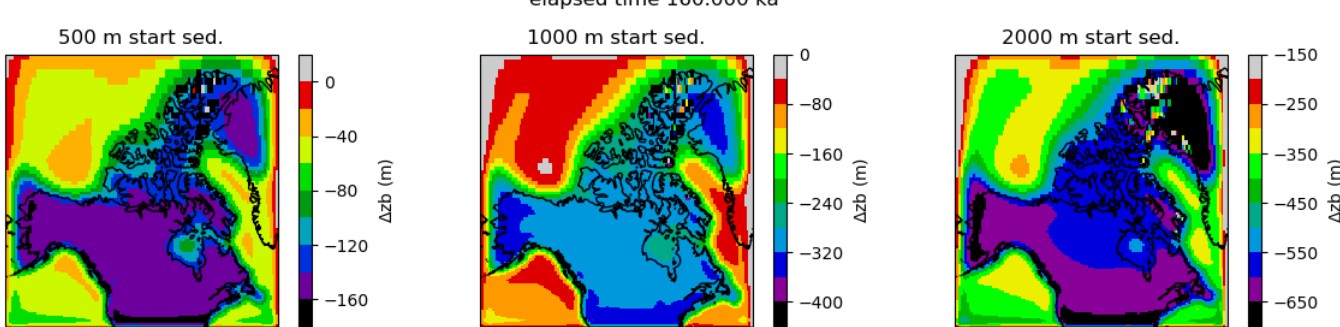

**Figure D1.** These isostatic adjustment tests show 40 kyr of isostatic adjustment (topographic effect of the sediment itself is removed) following sediment load emplacement. The load is imposed uniformly over the whole model domain: 500, 1000, and 2000 m.

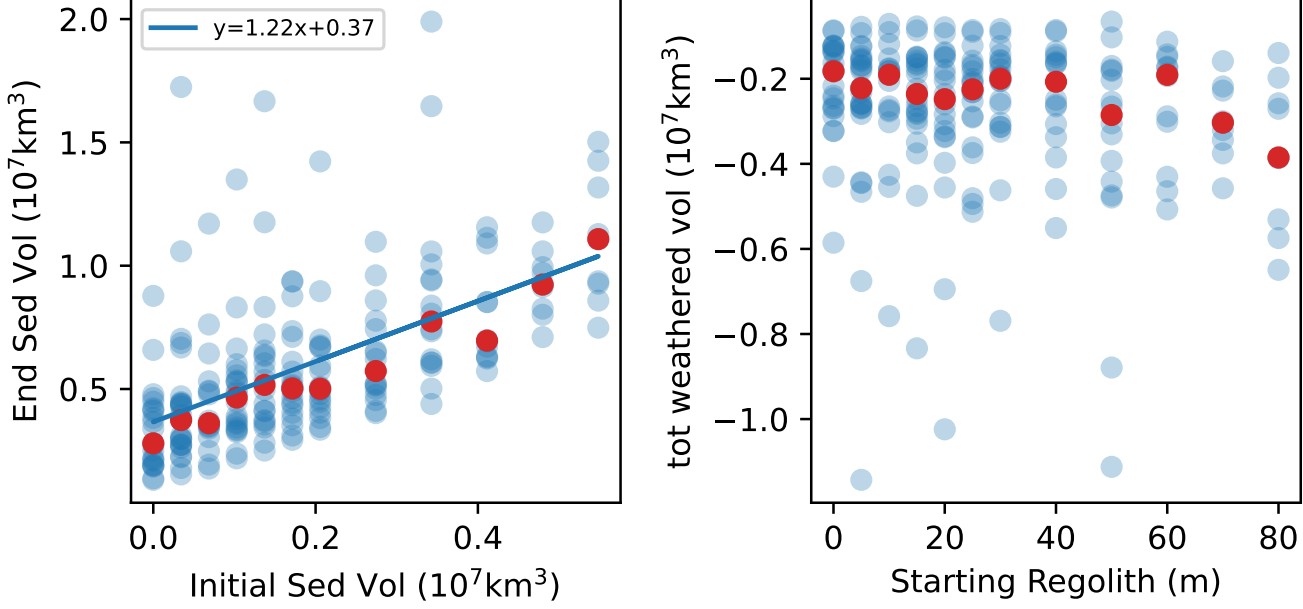

**Figure E1.** A comparison of the amount of sediment produced during glaciation with the amount of pre-existing regolith. The lefthand panel shows final sediment volume against initial sediment volume with a nearly one to one (1.22) linear slope – the final sediment volume is controlled mostly by input (initial) sediment volume. The righthand panel shows the total weathered (produced) volume of sediment against initial regolith thickness. The bulk volume of sediment produced is largely independent of the starting thickness. Around one to four million $km^3$ of sediment is produced throughout the Pleistocene – on the same order as the y-intercept in the lefthand plot.



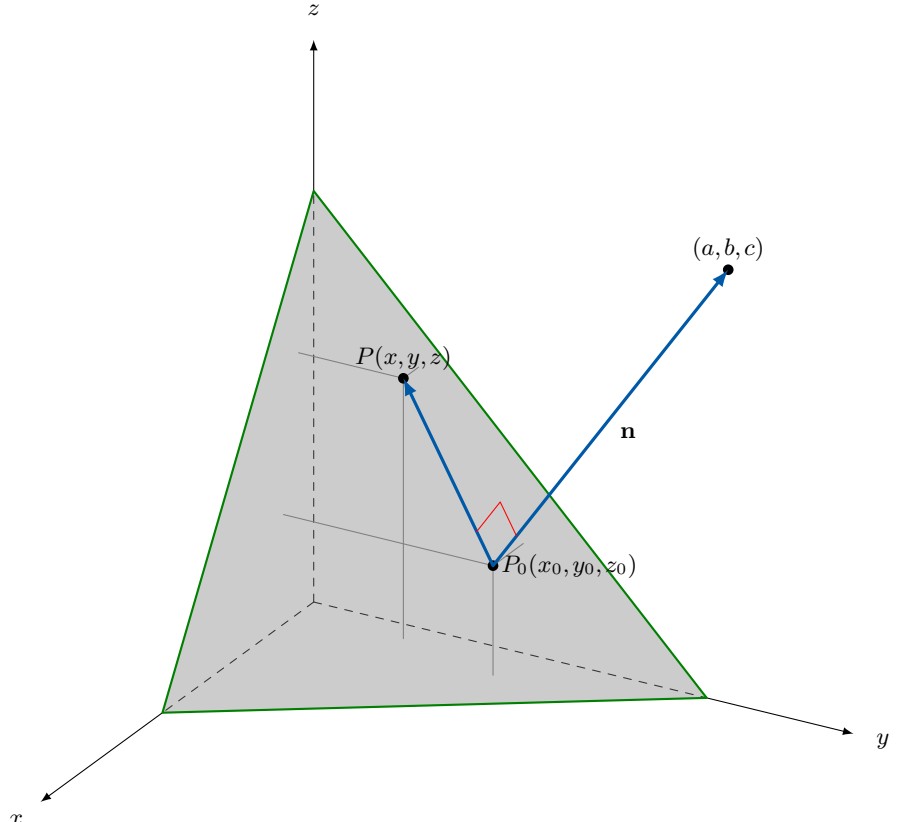

**Figure F1.** Illustration of calculating the normal vector to a plane from two non-colinear vectors lying within that plane.


$$P2P0 = P2 - P0 \tag{F2}$$

The normal vector to that plane can be expressed as the cross product of these two vectors,

$$P_{norm} = P1P0 \times P2P0 \tag{F3}$$

The dip angle between the approximated plane and the surface of the geoid is then

$$\theta_{dip} = arccos \frac{P_{norm} \cdot n_{geoid}}{\|P_{norm}\| \|n_{geoid}\|} \tag{F4}$$

In the NSIDC Sea Ice Polar Stereographic North projection the geoid normal unit vector has a z-component only. The dip angle then becomes:

$$\theta_{dip} = arccos \left[ \frac{P_{norm}}{\|P_{norm}\|} \right]_z \tag{F5}$$

Here we calculate these dip angles for both the covered and exposed bedrock regions of ArcticDEM coverage within onshore Canadian territory (shown in fig. F2). The two distributions are compared in fig. F3. We find negligible difference in distribution between these two domains with maximum likelihood dip angle around $0.55°$ and truncate the distribution at $10°$. This hard bed dip angle distribution is used to calculate the abrasion rates throughout this paper.




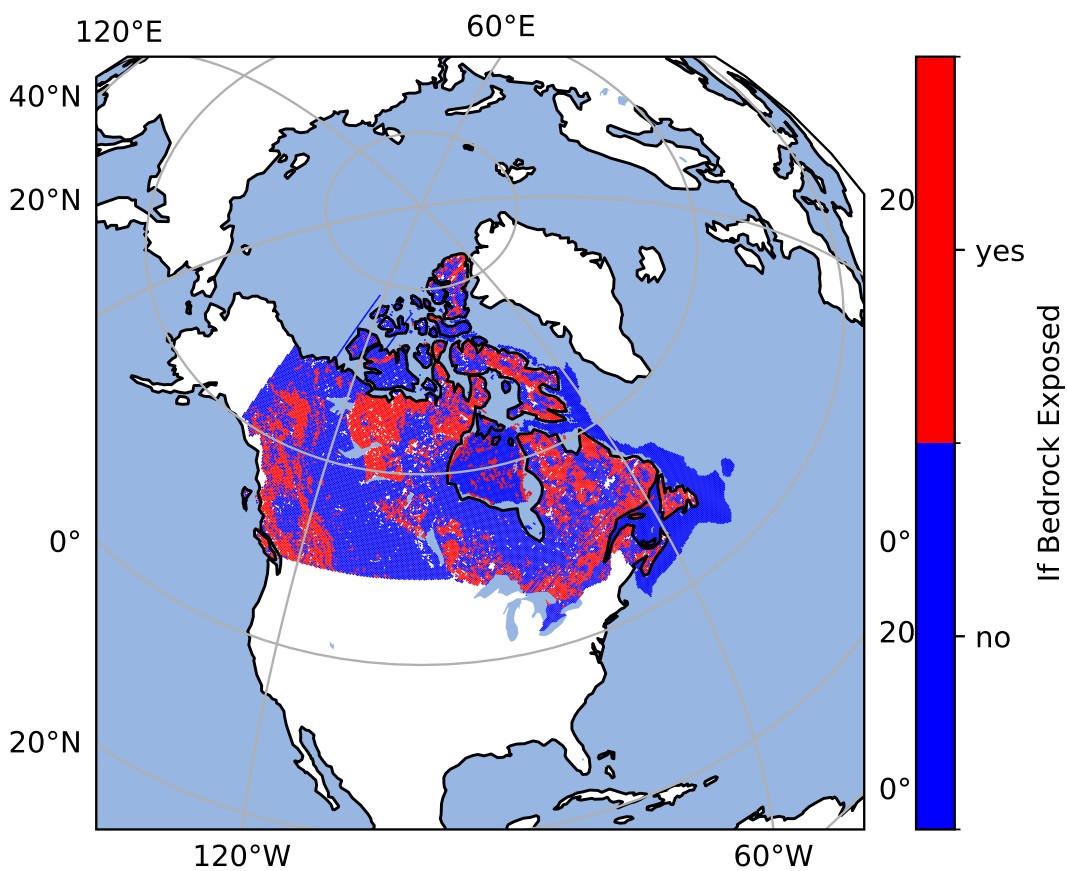

**Figure F2.** Map of exposed (hard, red) and covered (soft, blue) bedrock regions across Canada. Hard regions are those determined as either undifferentiated bedrock or till veneer in Fulton (1995), everything else is deemed soft (equivalently covered bedrock).



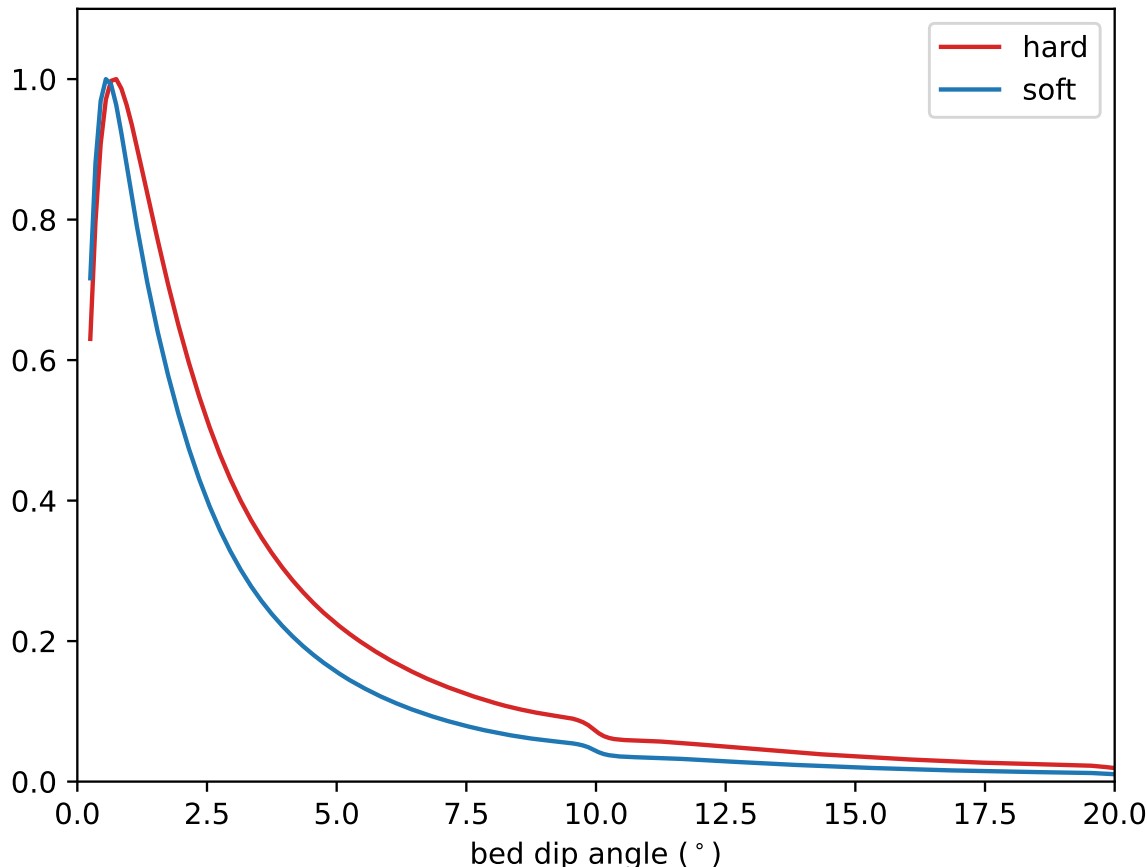

**Figure F3.** Dip angle distribution for hard and soft bedded regions shown in fig. F2.

*Author contributions.* MD performed sediment model development, devised/implemented the hydrology/sediment/ice dynamics coupling, experimental design & execution, analysis, and writing. LT maintains the GSM, devised/implemented the sediment/GIA coupling, advised experimental design, and edited the manuscript.

*Competing interests.* The authors declare no competing interests

*Acknowledgements.* We thank Stewart Jamieson for helpful discussion and review of an earlier version of this manuscript in his role as external Thesis examiner for MD.



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
