# Peer review of "North American Pleistocene Glacial Erosion and Thin Pliocene Regolith Thickness Inferred from Data-Constrained Fully Coupled Ice-Climate-Sediment modelling"

_EGUsphere, 2024_

## Referee Comment (RC3)

Review of: "North American Pleistocene Glacial Erosion and Thin Pliocene Regolith Thickness Inferred from Data-Constrained Fully Coupled Ice-Climate-Sediment modelling"

by Matthew Drew and Lev Tarasov

**General comments:**

The manuscript would benefit from a rewrite to enhance clarity and ensure consistent notation. For example, different notations are used for Table 2 and Figure 2, and numerous references are either unknown or sections are incomplete (such as Appendix E). Additionally, although Figure 6 is presented, it lacks reference, leaving its intended message unclear. These aspects collectively contribute to an impression of incompleteness within the manuscript.

In addition, a major point to me is that the focus of the manuscript is unclear to me. I do not understand which are the key variables of your model and how the coupling between your sediment-model and ice-sheet model works. The manuscript is very long and dense and difficult to read. I would consider splitting this manuscript in two manuscripts: one where the model physics and sensitivities are described more in detail with consistent notation (part A), and another where you apply it to the Laurentide domain (part B). Currently, the manuscript's structure is disorganized, with shifts between experiments in idealized and complex domains, as well as between present-day and paleo constraints, making it difficult for readers to follow.

I do not want to discourage the authors since I believe there is value behind this study but I think it needs to be reorganized to be understandable. Of course, since I am not an expert on this topic, part of the lack of knowledge could be related to my background, but the way the manuscript is written right now it feels unfinished and the figures are not clear.

**Main manuscript:**

1. **Introduction**

   ### 1.1 Current Models of Glacial Sedimentary Processes
   I do not understand the main outcomes of each sediment model described here. It becomes clearer when I read section 3, that is why I think part of Section 3 should be moved to the Introduction section.

2. **Model of glacial sedimentation**
   I think you should write here your main equations, not in the appendix. You should describe in detail the abrasion and quarrying process. It is written very descriptive and with many citations, but without any equation it is confusing for the reader. Also, something that is not clear to me is what are the main variables you are interested in. I understand that you have on one hand the sediment thickness and the sediment transport. These are then transferred into your ice-sheet model via the effective pressure or a friction coefficient? How do you compute the effective pressure?

   It would be very helpful if you could include a Figure in the form of a sketch where you summarize your main processes in your sediment model and how the ice-sheet model interacts with them like Quarrying, Abrasion, and Physical Weathering.

   **2.1 The Glacial Systems Model**
   It is unclear to me what climatologies you are using. Are you using a mean of all the PMIP ensembles or different climatologies? Maybe you could provide a snapshot of temperature and precipitation of the LGM state.
   Also, how does the presence of a Keewatin Ice Dome affect your results?

   **2.2 Sediment-Isostasy coupling**
   Is your IA in Equation (1) the same as your $L_{sed+eros}$ in equation (2)? If not, how do they relate?

   **2.3 Basal Drag Coupling**
   What values do you use for $c_{hard}$ and $c_{soft}$? How do you compute $f_{Neff}$?

   **2.4 Sediment Model Verification Tests**
   If your verification tests are shown in the Appendix, then you do not need to add this section in the main manuscript.

   **2.5 Sediment Transport Sensitivity for a Square Root Ice Sheet**
   This part is very confusing to me. In my opinion, if this is just a sensitivity analysis on an idealized domain (a square root ice sheet) then I would move this whole section to the Appendix. Also, Figure 2 is very difficult to read since your variable names are different from Table 2.

   Your Figure 2 shows a melt colorbar which is impossible to understand since you have not introduced any melt in the previous description. If I understand this correctly, you say that you test two sensitivity metrics to find your most

sensitive parameters. Then you say those parameters are " subglacial hydrology parameters, sediment viscosity, and the shielding factor". However, when I look at Figure 2, your shielding factor (ho) seems to me invariant. Maybe I am missing something here, but to me it is unclear.

**3. Constraints**

This section was nice to read but I am not sure it fits here. It feels a mixture between Introduction/Motivation and Discussion.

**4. Uncertainty in CO2 Forcing**

I think this should be part of the Appendix.

**5. Methodology**

For me, that the Methodology Section starts in Section 5 feels too late. Which parameters are you varying in your ice sheet model? If I understand correctly you first run an ensemble without hydrology nor sediments to constrain your dynamical parameters and then you  redo the same analysis with your sediment model? Your forcing comes from section 4?

How does your sediment model affect the simulated ice sheet?

**Appendix:**
**Appendix A:**
I think Appendix A should be a Supplementary Material rather than an Appendix. I do not understand the Units of the table and you do not refer to most of them.

**Appendix B:**
This appendix should be in the main manuscript in my opinion.
This part is very confusing to me. I see too many equations and it is not obvious how they relate to each other. For instance, you introduce an abrasion coefficient (L557: $C_{abr}^{*}$) which I do not see in any of your equations.

Equations here are difficult to read. Equation B35 is unreadable and it presents also an empty bracket.

**Appendix C:**
Your font size hereon becomes smaller and difficult to read.

**Appendix D:**
It is very difficult to understand your figure D1 if your color scales are different for every map.

**Appendix E:**
This part is empty, only a Figure without further description. You could include this as a Supplementary Material rather than an Appendix.

**Technical comments:**

When you refer to figures or equations you should use capital letters (Fig., Eq. and Table for example).

**Main manuscript:**
L26: What do you mean with "dubious physical reasoning"?
L28: "underlying glacial deposits (Setterholm and Morey, 1995)"
L31: what do you mean by "sparse glaciation above 1800 meters above sea level"?
L35: unknown reference.
L48: define in the introduction what you mean by "chemical dissolution"
L125: "... surface drainage solver of Tarasov and Peltier (2006),"
L164: unknown reference.
L165: unknown reference.
L235: "Interestingly, only Pelletier et al., (2016) gives ..."
L416: unknown reference.
L444: unknown reference.

**Appendix:**
L566: unknown reference.
L724: unknown reference.

- I think Figure B1 and C1 are repeated?